



# Assessing rainfall radar errors with an inverse stochastic modelling framework

Amy C. Green[1], Chris Kilsby[1], and András Bárdossy[2]

[1]School of Engineering, Newcastle University, Cassie Building, Newcastle upon Tyne,Tyne and Wear, NE1 7RU, United Kingdom
[2]Institute for Modelling Hydraulic and Environmental Systems, University of Stuttgart, Stuttgart D-70569

**Correspondence:** Amy C. Green (amy.green3@newcastle.ac.uk)

**Abstract.** Weather radar is a crucial tool in rainfall estimation, providing high-resolution estimates in both space and time. Despite this, radar rainfall estimates are subject to many error sources – including attenuation, ground clutter, beam blockage and the drop-size distribution – with the true rainfall field unknown. A flexible stochastic model for simulating errors relating to the radar rainfall estimation process is implemented, inverting standard weather radar processing methods, imposing path-integrated attenuation effects, a stochastic drop-size distribution field, along with sampling and random errors. This can provide realistic weather radar images, of which we know the true rainfall field, and the corrected 'best guess' rainfall field which would be obtained if they were observed in the real-world case. The structure of these errors is then investigated, with a focus on frequency and behaviour of 'rainfall shadows'. Half of simulated weather radar images have at least 3% of significant rainfall rates shadowed and 25% had at least 45km$^2$ containing rainfall shadows, resulting in underestimation of potential impacts of flooding. A model framework for investigating the behaviour of errors relating to the radar rainfall estimation process is demonstrated, with the flexible and efficient tool performing well at generating realistic weather radar images visually, for a large range of event types.

## 1 Introduction

Precipitation is extremely difficult to measure accurately, due to its intermittent nature, spatio-temporal variability, and sensitivity to environmental conditions (Savina et al., 2012). For urban hydrology weather radar plays an increasingly important role in quantitative precipitation estimation, due to the high spatio-temporal resolution of information needed (Thorndahl et al., 2017). The small size of urban catchments and the intended hydrological applications – particularly for real-time or near real-time – require information about precipitation fields at small temporal and spatial scales, from 1–10 minutes and 1–5km, respectively (Berne et al. 2004, Ochoa-Rodriguez et al. 2015, De Vos et al. 2017, Thorndahl et al. 2017, Shehu and Haberlandt 2021).

Despite the suitability of weather radar for obtaining high-resolution rainfall estimates, there are many sources of error in the estimation process, with different sources of uncertainty reviewed in numerous studies (Michelson et al. 2005, Meischner 2005, Villarini and Krajewski 2010, Ośródka et al. 2014, Ciach and Gebremichael 2020). Errors include radar calibration and stability problems; contamination by clutter and anomalous propagation; occultation; attenuation and assumptions made about the drop-size distribution (DSD) (Marshall and Palmer 1948, Harrison et al. 2000). Some error sources can be corrected for,



such as bias and systematic errors, ground clutter (Gabella and Notarpietro 2002, Ventura and Tabary 2013, Li et al. 2013) and attenuation (Nicol and Austin 2003, Krämer 2008, Jacobi and Heistermann 2016), resulting in significantly improved reliability. Correction procedures are often limited, due to the cumulative nature of errors from a superposition of different sources, with complex approaches showing only modest improvements to estimates. Information on the rainfall field is lost, irretrievable, and we do not even know how often this happens.

There is therefore an ongoing need to account for errors in the radar rainfall estimation process (Villarini and Krajewski 2010, Seo et al. 2018) and uncertainties should be acknowledged and modelled (Ciach et al. 2007, Gires et al. 2012, Villarini et al. 2014, Rico-Ramirez et al. 2015). The poor quantification of uncertainties was highlighted as a fundamental issue in AghaKouchak et al. (2010a), expanded in AghaKouchak et al. (2010b). An error model described in Hasan et al. (2014) found uncertainties were easily identifiable for unbiased $ZR$-relationships, incorporating radar reflectivity uncertainties in Hasan

et al. (2016). Variograms were used to representing radar rainfall uncertainties Cecinati et al. (2017), eliminating the need for a covariance matrix for faster and more flexible calculation of the spatial correlation of errors. Uijlenhoet and Berne (2008) created a stochastic model of range profiles for the DSD, using a Monte Carlo framework (Berne and Uijlenhoet, 2006) to estimate uncertainties using two attenuation correction schemes. Yan et al. (2021) imposed random and non-linear radar errors on simulated rainfall fields, with $ZR$-relationship errors appearing to have little influence overall.

Error quantification is challenging and errors propagate into future estimates for any model which requires rainfall as an input. The fundamental limitation in radar correction that the 'true' rainfall field is not available for comparisons. In this study, the aim is to work backwards to obtain an estimate of the uncertainty in the radar rainfall estimation process. Using a new model for simulating realistic space-time rainfall event fields with a high resolution (matching that of a U.K. standard C-band weather radar) (Green et al., 2023), clustered parametrisation based on radar rainfall events extracted from the U.K. Met Office

operated High Moorsley weather radar. These simulation outputs are treated as the 'true' rainfall field. Errors relating to each step of the radar rainfall estimation process are then imposed on the simulated rainfall field, to obtain an ensemble of spatio-temporal error fields for each event, in a stochastic manner, forming a superposition of different error sources. This is done by inverting standard radar processing methods, allowing the identification of the frequency of occurrence and extent of the loss of important information.

In this study, the data and study area are first discussed, as well as the simulation methods applied to obtain realistic space-time rainfall fields in Sect. 2. The methodology for the radar error model is then outlined in detail in Section 3, with detailed explanations for each step of the model. Example event results are discussed in Section **??**, with more general results based on event images given in 5. A discussion and conclusions are given in Section 6, with model limitations, potential for generalisation and future work also discussed.

## 2   Data

An ensemble of realistic rainfall events are used, generated using the clustered rainfall model outlined in Green et al. (2023). This model uses Fast Fourier Transform (FFT) methods to efficiently generate three dimensional rainfall event fields with a




high-resolution matching that of radar data (1km, 5min) for a $200 \times 200$km domain. Events have prescribed properties, includ-

ing the correlation structure, spatial anisotropy, spatio-temporal anisotropy, marginal distribution, non-zero rainfall proportions

and advection. The model is used with multidimensional scaling and hierarchical clustering to parametrise rainfall event sim-

ulations, for 100 rainfall events. A year of processed dual-polarisation C-band weather radar data, from the High Moorsley

weather radar (north east England) as part of the Met Office radar network is used. This dataset is used to parametrise simula-

tions of realistic space-time rainfall fields.

## 3 Radar error model

This section outlines a novel model for imposing errors in the radar rainfall estimation process on a rainfall field, focusing

on four main error sources: random noise effects, attenuation effects, DSD error and sampling through estimation variance.

Sections 3.1–3.3 describe the error model in more detail, outlined in Fig. 1, written in Python. While the model is by no means

comprehensive, random error is included in the model. This is designed to provide a framework for investigating the impact of

these errors, improving understanding of the estimation process.

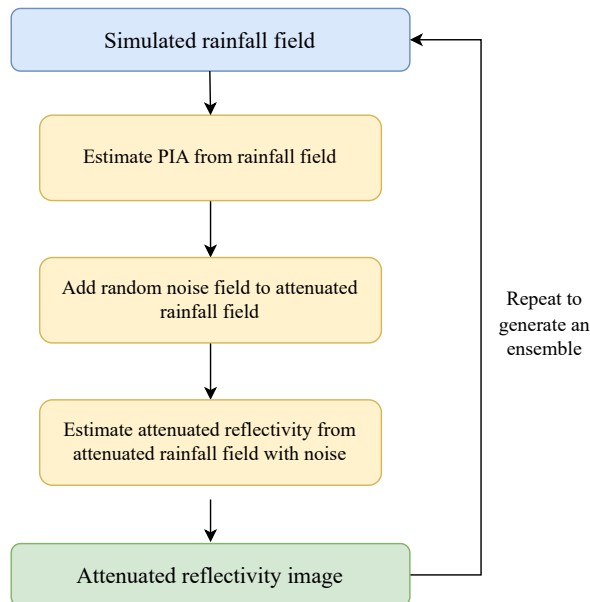

**Figure 1.** Schematic for imposing the radar error model on simulated rainfall fields

## 3.1 Reprojecting to polar coordinates

The simulated rainfall fields are given on a regular three-dimensional Cartesian grid. To apply radar processing methods

in reverse, the data must be reprojected into a polar coordinate system. Using nearest neighbour interpolation methods, the



Cartesian grid is converted into polar data

$$Z(t,x,y) \rightarrow Z(t,\theta,r) \tag{1}$$

for ray angles $\theta = 1, 2, \dots 360$ with ray bins $r = 1, 2, \dots 167$ of width 600m and average elevation angle of 1 degree. This mirrors the radar configuration of the High Moorsley weather radar, used for parametrisation. The different elevation angles and difference in sampling sizes of pixels are incorporated through the use of estimation variance in Sect. 3.5.

## 3.2 Attenuation effects

A constrained gate-by-gate approach is applied to estimate the path-integrated attenuation (PIA) for each radar ray by inverting standard forward attenuation models (Krämer and Verworn 2008, Jacobi and Heistermann 2016). Inverting the process gives an estimated attenuated reflectivity $Z_i$ rate for the $i$th bin of width $\Delta r$ as

$$\hat{Z}_i = Z_{i,corr} - \sum_{j=0}^{i-1} \hat{k}_j \qquad \hat{k}_i = c \left[ Z_{corr,i} + (2\Delta r - 1) \sum_{j=0}^{i-1} \hat{k}_j \right]^d \tag{2}$$

for constants $c$ and $d$. This results in a realistic radar image of attenuated reflectivity in a polar coordinate system at each time step of the event, denoted by $Z_{corr}(t,\theta,r)$. Using the scheme described above, we get a PIA estimate $PIA(t)$ of

$$\tilde{R}(t) = \begin{cases} R(t) - PIA_{R(t)}(t), & \text{if } R(t) \geq PIA_{R(t)}(t) \\ 0, & \text{otherwise.} \end{cases} \tag{3}$$

$$\tag{4}$$

where $PIA(t) = f\{R(t)\}$ is a function based on the estimation algorithm outlined in Jacobi and Heistermann (2016).

## 3.3 Random noise effects

When considering empirical variograms for weather radar images, Pegram and Clothier (2001) found 10% of the variability in images corresponded to nugget effects, highlighting the need for random noise effects in radar pixel simulations. This noise is also evident in the marginal distributions of radar images, with the full year and an example 'dry' day image for the High Moorsley weather radar given in Fig. 2, showing a large number of values in the range $-32$–$0$dBZ. Although rainfall rates of less than $0.01$mmh$^{-1}$ are hardly noticeable in terms of rainfall accumulations, this high density of low reflectivity rates in radar images may have a significant effect on attenuation estimates along the radar rays. This noise may be attributed to the measuring apparatus, non-meteorological echoes, or most likely a combination of various different sources. Errors are treated as random noise, representing a combination of errors from unknown sources, clearly evident in real radar images.

The random noise field is added to rainfall values to prevent instabilities, with the marginal distribution from Fig. 2 converted to rainfall rates in Fig. 3. A log-normal marginal distribution would allow for a simple and easy transformation when simulating the field using Gaussian random field theory. Empirical variograms of these values were estimated to identify an appropriate



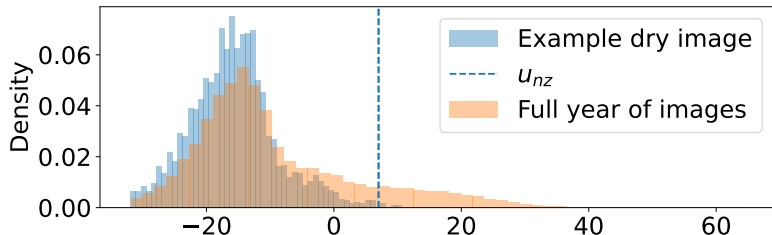

**Figure 2.** Histograms of an example dry radar image (blue) and all radar pixels above $-32$dBZ for the High Moorsley weather radar for the year 2019, where $u_{nz}$ is the reflectivity rate corresponding to the non-zero rainfall threshold of $0.1$mmh$^{-1}$

correlation structure, which has a very short correlation range of around 5km, with no clear anisotropy effects. The three-dimensional noise field denoted by $\varepsilon(t, x, y)$ is assumed to be log-normal, with a marginal distribution of

$$\varepsilon \sim LN(\mu_\varepsilon, \Sigma_\varepsilon^2) \tag{5}$$

where $\mu_\varepsilon = -5.3$ and $\sigma_\varepsilon = 1.7$. A Gaussian random field is simulated with an Exponential correlation structure of $\rho_\varepsilon(h) = \exp\{-h/r_\varepsilon\}$ for a short range with $r_\varepsilon = 5$ and a nugget effect of $n_\varepsilon = 0.35$. This is transformed using an inverse Gaussian score transformation, then exponentiated resulting in a random noise field with the desired marginal distribution and correlation structure. An example field is included in Fig. 3, from which we can see that the variability is slightly larger than in existing images. This is however selected to preserve the proportion of $-32$dBZ reflectivity rates in images, with any values less than $-32$dBZ treated as $-32$dBZ.

## 3.4 Drop-size distribution errors

Attenuated rainfall rates $\tilde{R}(t)$ can then be added to the three-dimensional noise field $\varepsilon(t, x, y)$, which can then be converted into a reflectivity field. A $ZR$-relationship is typically used, of the form $Z = 10\log_{10}(aR^b)$ for reflectivity $Z$ (dBZ), rainfall $R$ (mmh$^{-1}$) and constants $a$ and $b$, which typically take the values $a = 200$ and $b = 1.6$ (Harrison et al., 2000). A constant value for $a$ and $b$ is based on the assumption that the DSD varies spatially and temporally in a way characteristic of a particular rainfall or weather type. Despite this, a fixed $ZR$-relationship results in a severe underestimation of peak rainfall intensities, due to the failure to account for natural variations in the DSD with intensity (Schleiss et al., 2020). Lee et al. (2007) indicated that the overall DSD variability cannot be adequately explained by a single parameter. In Libertino et al. (2015), a varying $ZR$-relationship in space and time improved rainfall accumulations at event scale, when comparing to a fixed relationship.

A large amount of scatter around the average power-law relationship is related to the various microphysical processes that are responsible for the DSD variability. To account for this variability, in an attempt to generate more realistic reflectivity images, we assume that $a = A(x, y)$ is a two-dimensional field varying in space. As the simulated rainfall events all have a fairly short duration (6 hours or less), a constant DSD in time is initially used. This assumes that $A$ is fairly constant over the time period considered, although the model is flexible and the dimensions of $A$ can be easily extended to include time.





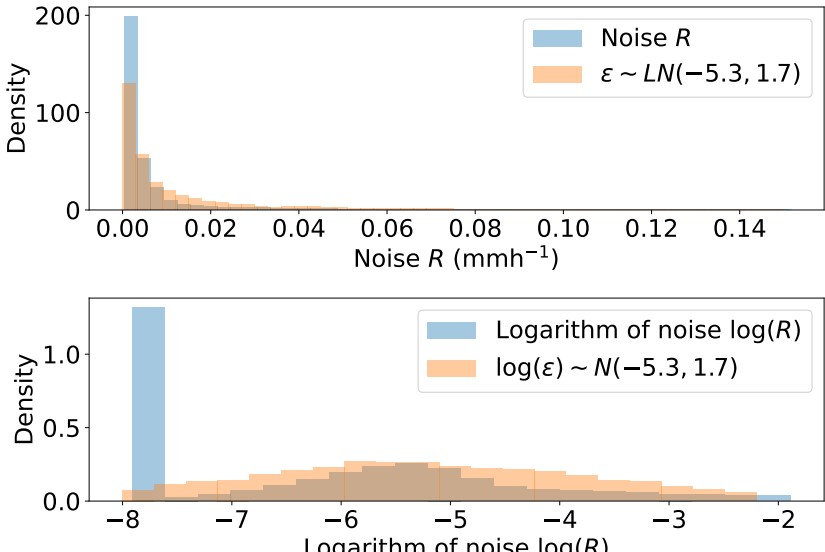

**Figure 3.** Histograms of low rainfall rates (and logarithms) corresponding to random noise (i.e. in the range $-32 < Z \leq 10$,) from High Moorsley weather radar for the year 2019, as well as an example simulated noise field

Parameters in the $ZR$-relationship typically take values in the ranges $a \in (30, 1000)$ and $b \in (0.8, 2)$ (Battan and Theiss 1973, Smith and Krajewski 1993), and so the parameter $b$ is still treated as constant, but sampled from a Gaussian distribution with a low variance, centred around a value of $\mu_b = 1.6$. This gives attenuated reflectivity estimates of

$$\tilde{Z}(t) = 10\log_{10}\left\{ A \left[ \tilde{R}(t) + \varepsilon(t) \right]^b \right\} \tag{6}$$

for

$$A(x,y) \sim N_2(\mu_a, \{1 + g(x,y)\}\Sigma_a) \qquad b \sim N(\mu_b, \sigma_b^2) \tag{7}$$

for correlation structure $\rho_a = \sigma_a \exp\{-h/r_a\}$ where $\mu_a = 220$, $\sigma_a = 2$, $r_a = 30$, $\mu_b = 1.6$, $\sigma_b = 0.02$. The function $g(x,y) = \Sigma_E(x,y)$ is the estimation variance based on pixel location $(x,y)$, based on the proportion of the rainfall volume that the radar can see for a given distance, which is discussed further in Sect. 3.5. Attenuated reflectivity fields are rounded to one decimal place, and limited by a minimum value of $-32$dBZ, as is the case for actual reflectivity data.

## 3.5 Radar sampling

Due to the nature of weather radar sampling, polar observations close to the weather radar location sample from a much smaller volume than those further away (see Fig. 4). As observations are made further from the radar, effects of curvature of the earth and the cloud height also affect the sampling volume of the radar. Bright band effects also impact the sampling volume, as



radar observations above the freezing level are often unavailable due to the high reflectivity of melting precipitation causing
strong returns at the bright band level (Hooper and Kippax 1950, Kitchen et al. 1994, Hall et al. 2015).

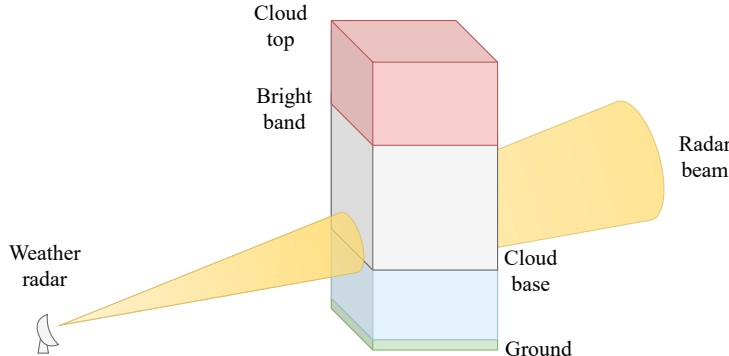

**Figure 4.** Schematic of a radar ray sampling volume for an example pixel, denoted as a vertical columns

Estimation variance principles are included in the DSD error to account for these sampling errors. The simulated rainfall field is on a regular grid (1km resolution pixels). This is assumed to represent the total rainfall in the vertical column of space above this kilometre square, where rainfall is falling on the ground. Making assumptions about the vertical behaviour of the DSD of rainfall, this is incorporated into the model through the variance of the multiplicative parameter in the $ZR$-relationship,
dependent on the DSD and freezing level height.

Representing the change in uncertainty based on the volume of rainfall sampled by the radar apparatus beam in a given pixel (at different distances from the radar location), the estimation variance is defined as

$$\sigma_E^2 = Var \left\{ \int_0^c Z(x,y,u) \left| \int_{h_{x,y}}^c Z(x,y,u)du \right. \right\} \tag{8}$$

where $h_{x,y}$ is the lower elevation of the radar scan at $(x,y)$. Simplifying to consider the two-dimensional case, heights are discretised from the ground to the cloud top and clipping at the freezing level (Hooper and Kippax, 1950), giving

$$\sigma_E^2 = \bar{\gamma}(V) - \sum_i^{n_r} \bar{\gamma}(v_i) \tag{9}$$

for distance from the radar $h$, vertical column of rainfall $V$, with radar ray sampling volumes $v_i$ for radar ray $i = 1, 2, \ldots n_r$, where $\bar{\gamma}$ represents the mean variogram, for $n_r$ radar elevations. For distances of the discretised volumes the estimation variance
is calculated using the variogram model corresponding to the assumed distribution. Parametrisation is based on vertical weather radar data, considering the vertical raindrop volume distribution for a range of rainfall rates and heights. The variance of sampling volumes at different ranges supported the concept, with simulations compared to vertical radar images in Berne et al. (2005) for validation.



single time step towards the middle of the event is included, with a link to event videos included in each figure.

Crane (1979) referred to distortions in storm structures, as a result of attenuation, as shadows. We formally define a rainfall shadow, taken as pixels where the simulated rainfall is significant (i.e. $R > 1\text{mmh}^{-1}$), but the corrected rainfall much lower (less than 10%) of the original simulated rate (i.e. $R_{corr}/R \leq 0.1$).

## 4.1 Example fields

For an example time step of a simulated event, each stage of the radar error model process is given in Fig. 5. The final radar image appears visually realistic, with clear areas of rainfall, similar to raw radar images obtained from the High Moorsley weather radar. A significant proportion of the signal is attenuated towards the edge of the domain, particularly in the top-right of the image.

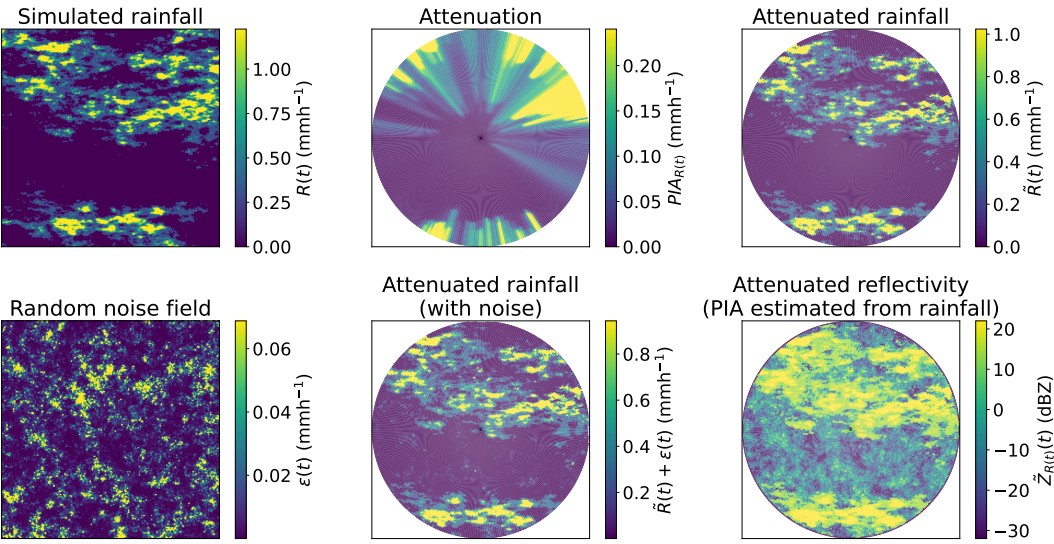

**Figure 5.** Step-by-step error model fields, including the simulated rainfall field $R(t)$, attenuation $PIA$, attenuated rainfall $\tilde{R}(t)$, random noise field $\varepsilon(t)$, attenuated rainfall with random noise $\tilde{R}(t) + \varepsilon(t)$, and attenuated reflectivity $\tilde{Z}(t)$ for a single time step of an example simulated event

## 4.2 Event A: High bias

The event shown in Fig. 6 has an area of moderate intensity rainfall in the centre of the image with a large extent, resulting in high bias. The simulated radar image for an ensemble member associated with the event looks realistic, with the reflectivity and corrected rainfall rates showing significant rainfall amounts missing throughout. The average bias, RMSE and pixel variability corresponding to the event in Fig. 7. The average bias and RMSE are very high, taking values over $5\text{mmh}^{-1}$.




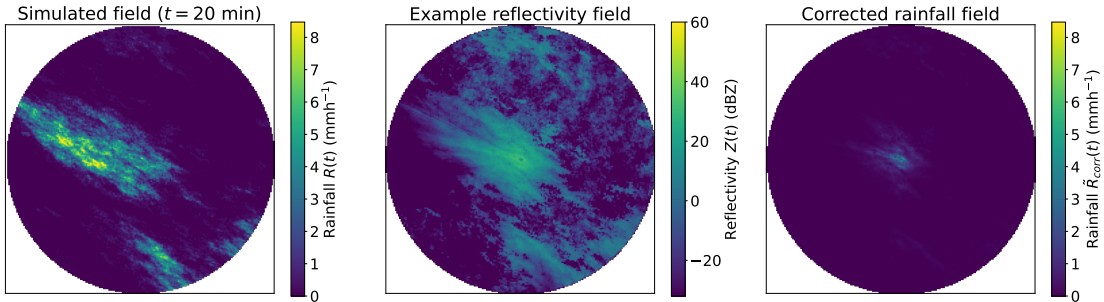

**Figure 6.** Simulated rainfall field and example reflectivity and corrected rainfall field for simulated event A (video link)

Pixel variability is very low for most of the image, except centre of top right minimum RMSE is greater than 5mmh$^{-1}$ in a large area of the domain, suggesting that the rainfall is consistently underestimated throughout the entire ensemble. A large area of moderate-intensity rainfall on top of the radar is overcorrected, mimicking effects resulting from full attenuation of the radar signal by intervening rainfall. In this case, the correction techniques could not improve the image significantly, and so

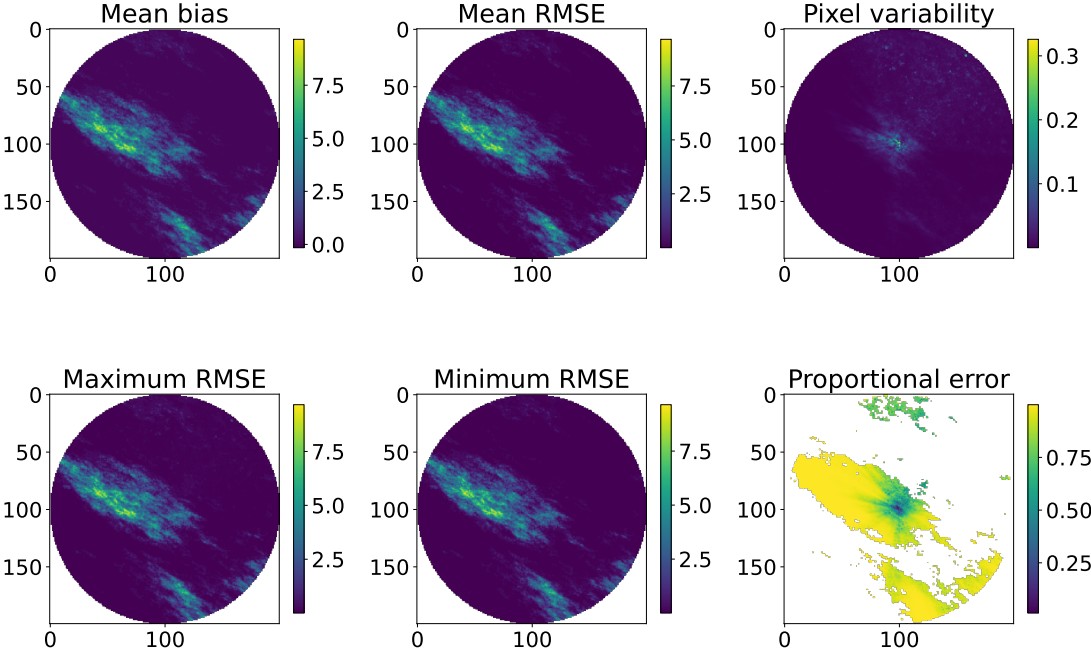

**Figure 7.** Average bias, RMSE, pixel standard deviation, maximum RMSE, minimum RMSE and average proportional error for simulated event A (video link)





information on a large portion of the rainfall field has been lost, particularly when forward attenuation correction algorithms
are implemented.

This result is reiterated when looking at the rainfall shadows in Fig. 8, where around a quarter of the image is shadowed, for
100% of ensemble members. This event has very high average bias, with pixel variability varying drastically throughout the
image. Large areas of rainfall are missing, and the differing variability throughout and spatial distribution of the error structure
suggests that a mean field bias or multiplicative correction would not improve estimates significantly. The information on the
rainfall structure are rates would be lost in this case.

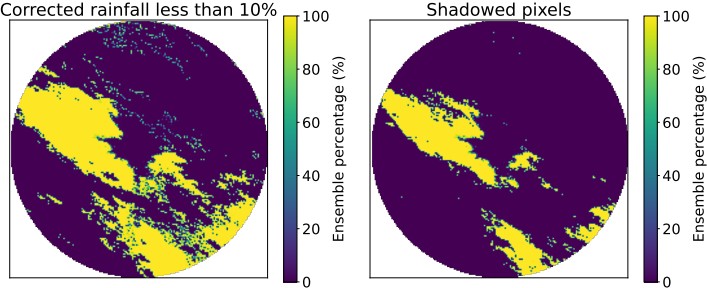

**Figure 8.** Percentage of corrected rainfall less than 10% of original simulated field, and frequency of shadowed pixels over the ensemble for
simulated event A (video link)

### 4.3 Event B: Low variability

Figure 9 shows a rainfall event with a small extent of light rainfall, with mostly mist and no rain throughout the image, resulting
in low variability. There is a small amount of light rainfall on the centre-left of the radar domain, with the corrected rainfall
image exhibiting lower rainfall rates here. The radar image appears realistic, with a small amount of signal damping towards
the left of the image, at a range beyond the light rainfall.

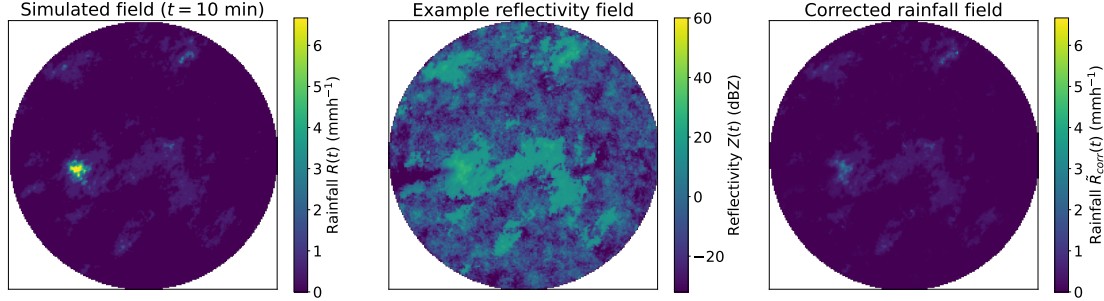

**Figure 9.** Simulated rainfall field and example reflectivity and corrected rainfall field for simulated event B (video link)





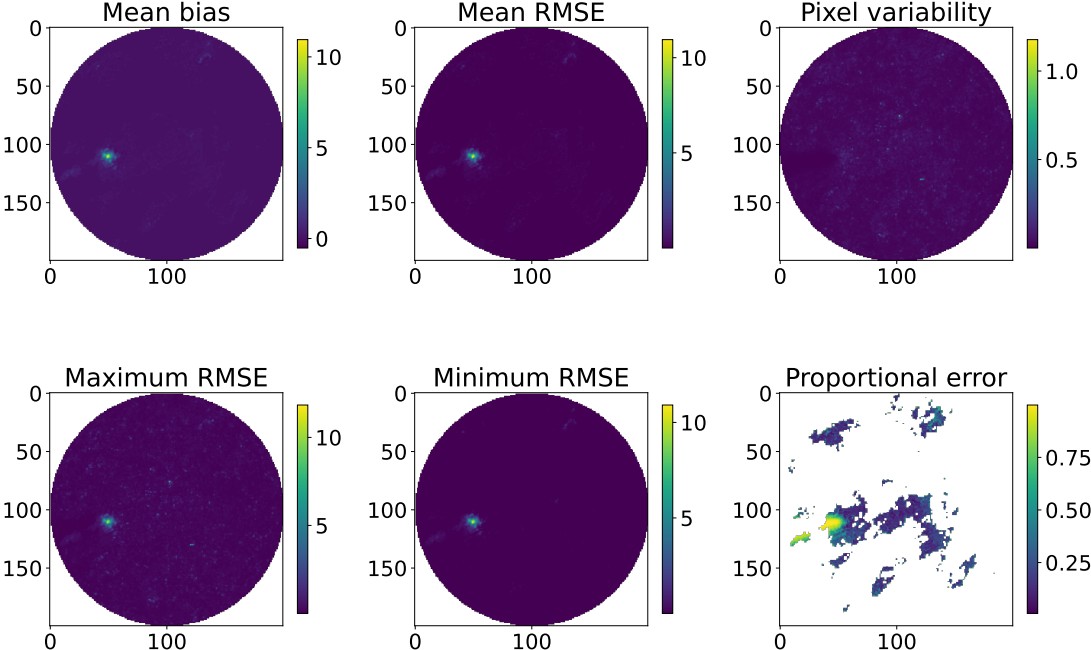

**Figure 10.** Average bias, RMSE, pixel standard deviation, maximum RMSE, minimum RMSE and average proportional error for simulated event B (video link)

The corresponding bias and RMSE for this event are given in Fig. 10, as well as the pixel variability, maximum and minimum RMSE over the ensemble and average proportional error. Over the ensemble, the average bias is close to zero except for the low-intensity rainfall areas (at most $0.5\text{mmh}^{-1}$), with low average, minimum and maximum RMSE. The pixel variability is slightly

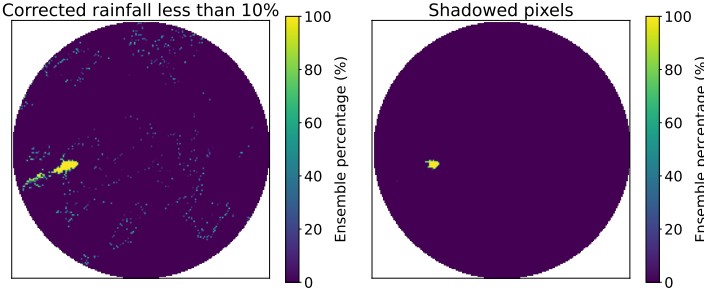

**Figure 11.** Percentage of corrected rainfall less than 10% of original simulated field, and frequency of shadowed pixels over the ensemble for simulated event B (video link)

 

higher in the rainfall area, with pixels radially further away in this area showing lower pixel variability (less than $0.02\text{mmh}^{-1}$),
with the remaining variability appearing uniform. In Fig. 11, the rainfall is shadowed in 100% of rainfall ensemble (i.e. all
ensemble members) in the area of light-intensity rainfall identified in Fig. 9. The frequency of shadows over the ensemble
taking mostly values of either zero or one. This event has very low variability between ensemble members, likely due to
(mostly) non-zero rainfall amounts in the images.

### 4.4 Event C: High variability

Figure 12 shows an event with a small area of heavy rainfall rates, which results in high variability in event errors. Most
of the radar domain shows zero rainfall rates, except a very small area of high-intensity rainfall (greater than $100\text{mmh}^{-1}$)
towards the top of the domain. The example radar image is again realistic, showing mostly noise. Radial lines at top right past
a small amount of high-intensity rainfall suggest that attenuation effects have not been sufficiently corrected.The corrected
rainfall image overestimates areas of high-intensity rainfall, due to cumulative errors introduced as part of forward attenuation
correction procedures. Although there is not a large area of high-intensity rainfall, the rainfall field spatial distribution has still
been significantly impacted by the errors caused by attenuation.

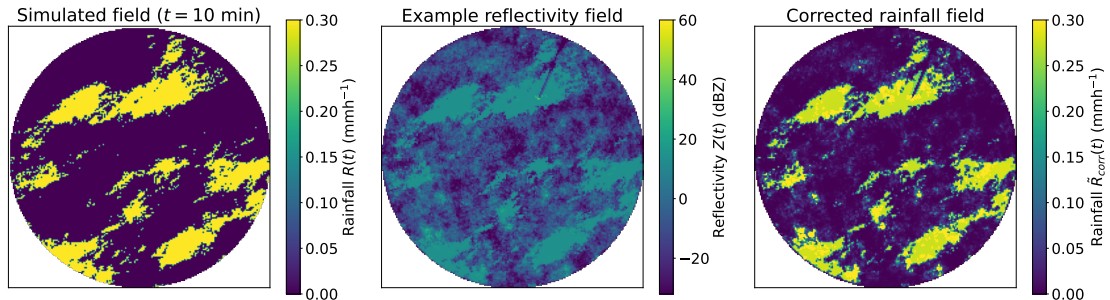

**Figure 12.** Simulated rainfall field and example reflectivity and corrected rainfall field for simulated event C (video link)

From Fig. 13, these radial lines show positive average bias and a higher RMSE than the rest of the image which is close
to zero, however none of these exceed $0.2\text{mmh}^{-1}$. Errors are not noticeable from the maximum RMSE field, but have much
higher minimum RMSE than the rest of the image. In this case, it appears that the attenuated high-intensity rainfall has been
over-corrected, with the area showing an average proportional error greater than one. Areas radially past this are underestimated
and although most of the pixels past this are light rainfall or mist, the rays have been significantly impacted, showing a clear
gap resulting from corresponding rainfall in the reflectivity images.

The shadowed pixels in Fig. 14 shows radial lines in areas where the corrected rainfall is less than 10% of the simulated
rainfall, most of these pixels do not have significant rainfall rates so are not classed as shadowed, with only a handful of pixels
showing shadows and none for 100% of the ensemble, resulting in high variability within the ensemble.

From the videos of event errors, this high-intensity rainfall moves across the image, affecting different rays, with some
pixels showing rainfall shadows past the high-intensity rainfall. This is not consistent throughout the ensemble, with most



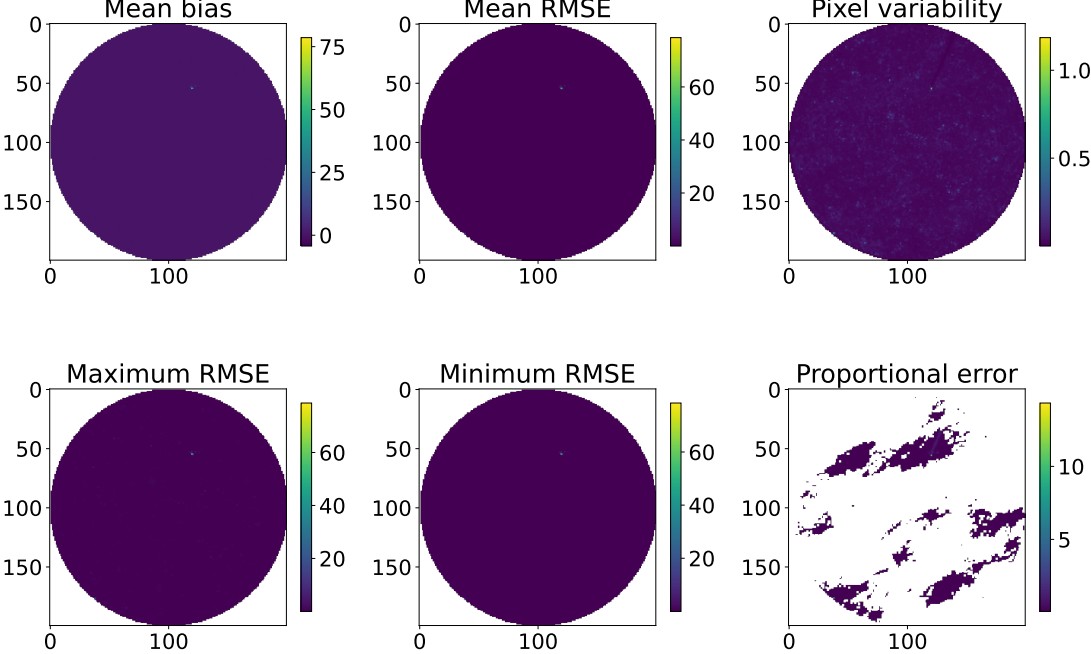

**Figure 13.** Average bias, RMSE, pixel standard deviation, maximum RMSE, minimum RMSE and average proportional error for simulated event C (video link)

pixels showing shadows in less than 100% of the ensemble members. This small area of high-intensity rainfall has resulted in high variability across the ensemble, which is impacted significantly by the over-estimation of high-intensity rainfall rates, with the DSD error also contributing towards the variability for such a high rainfall rate.

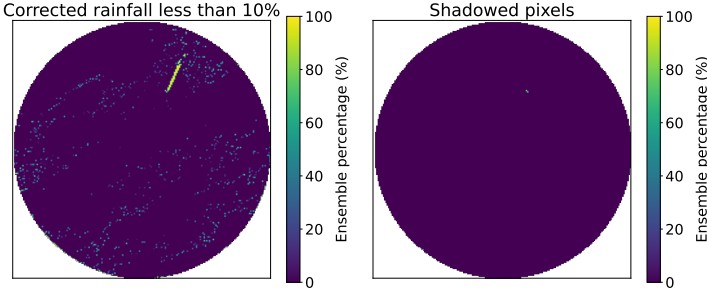

**Figure 14.** Percentage of corrected rainfall less than 10% of original simulated field, and frequency of shadowed pixels over the ensemble for simulated event C (video link)





## 4.5   Summary

In cases where the absolute bias is low, rainfall shadows may still exist, suggesting average bias is not a good metric to use when identifying event errors. Even with a low absolute bias, the rainfall could be overestimated for pixels closer to the radar, and underestimated past these pixels (which is very common along radar rays where attenuation has been overestimated). In these cases, the spatial distribution of rainfall is often incorrect, which could have detrimental effects when using the rainfall fields for any quantitative modelling.

Typically events with high bias correspond to events with large rainfall shadows, with shadow frequencies close to either zero or one, suggesting that in cases where there is high bias, the model uncertainty is low. Simulated rainfall events which exhibited high ensemble variability were as a result of potential interactions between small areas of high-intensity rainfall and DSD errors, which corresponded to higher variability in shadows throughout the ensemble. Videos of all simulated events (and corresponding errors) are available here.

## 5   Results: Individual image-based errors

The behaviour of individual radar images is considered, looking at the average, minimum and maximum behaviours over the ensemble, including the variability. In this section, attempts are made to find metrics and properties of rainfall images with the aim of identifying instances where there is a very high level of uncertainty or error arising from the rainfall estimation process. The impact of the rainfall location with respect to the radar is considered, as well identifying how often significant information on the rainfall field is lost. For individual image based errors, we introduce three image metrics relating to rainfall shadows given below.

1. ARS: The actual area (km$^2$) of the radar image that contains rainfall shadows.

2. LARS: The largest single area (km$^2$) of rainfall shadows in a radar image.

3. PRS: The proportion of significant rainfall (i.e. $R > 1\text{mmh}^{-1}$) that is shadowed.

### 5.1   Average ensemble behaviour

The relationship between the average rainfall rate and proportion of rainfall in an images with the image RMSE is shown in Fig. 15, showing that events with high average rainfall rates and large heavy rainfall proportions have the highest RMSE. Images showing fairly low proportions (i.e. 5–10%) of heavy rainfall still exhibiting fairly high RMSE. Figure 15 also shows the relationship between the mean and standard deviation of non-zero rainfall rates with the image RMSE, showing higher RMSE for events with high average and standard deviation in non-zero rainfall rates. This may be due to the large errors resulting in large gradients between pixels, where a large rainfall rate along a ray damps the signal so that subsequent observations are much lower, increasing pixel variability in image.



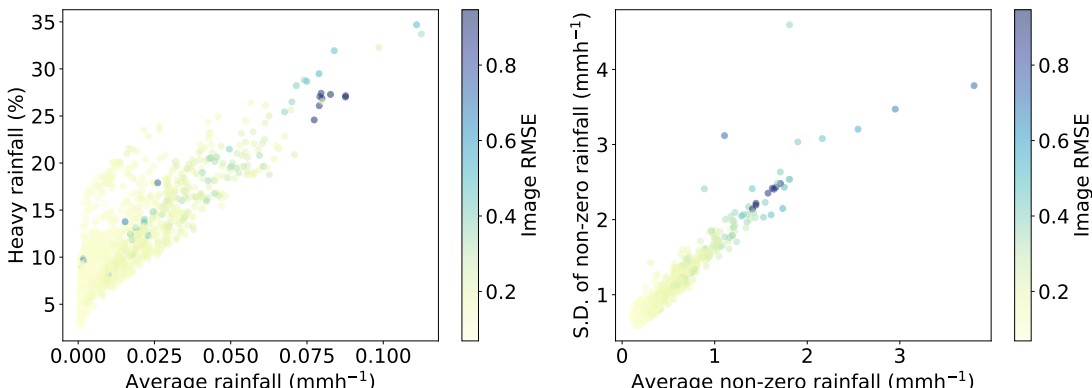

**Figure 15.** Average rainfall rate and proportion of heavy rainfall (left) and average non-zero rainfall and standard deviation of non-zero rainfall (right), coloured according to the image RMSE for event images

Figure 16 shows the average rainfall rate and proportion of non-zero rainfall in event images, for the average bias, RMSE, ARS and PRS. For larger rainfall rates and proportions of rainfall, the bias increases, with low proportions of low rainfall rates exhibiting negative average bias. This is also the case for the average non-zero rainfall rates, however the relationship between the proportion of non-zero rainfall and RMSE is less distinct, highlighting the significant impact of very small areas of intense rainfall rates on the image RMSE. Figure 16 also shows that there are some low average rainfall rates and proportions of non-zero rainfall which correspond to high areas of rainfall shadowed. This may be attributed to noise, however it does suggest again that events do not need to include intense large-scale rainfall areas to result in significant rainfall shadows. The

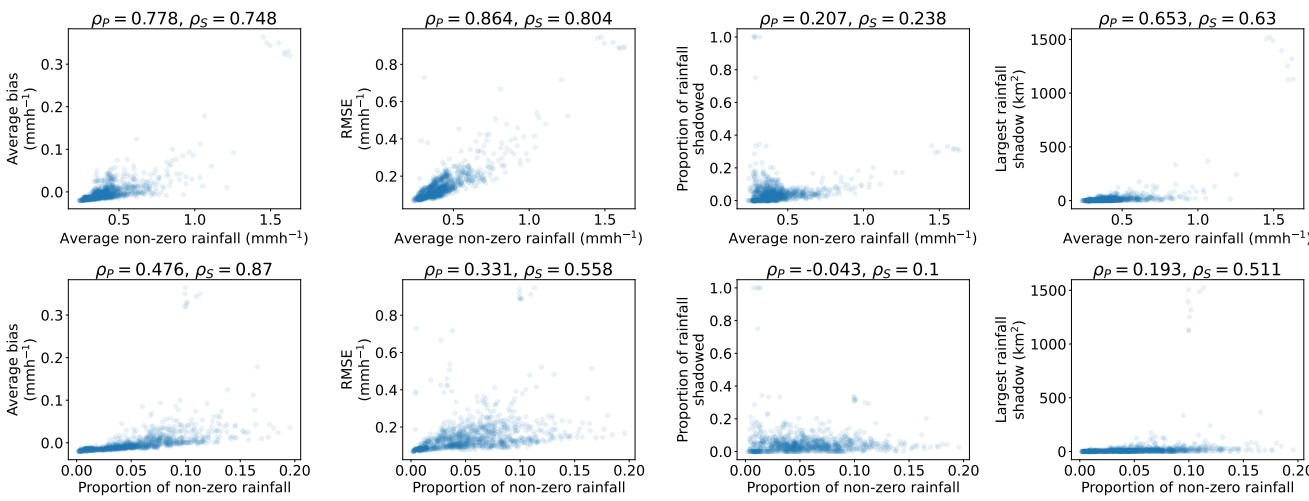

**Figure 16.** Average non-zero rainfall rate and proportion of non-zero rainfall and average bias for ensemble images of all events





proportion of rainfall shadowed also appears to increase exponentially as the average non-zero rainfall rate increases, which is not the case with the proportion of rainfall in images.

The relationships in Fig. 16 are heavily skewed by a high density of images with low average rainfall rates and proportions.
For a clearer image of the behaviour for event images, we consider images with an average rainfall rate of at least $0.1\text{mmh}^{-1}$ in Fig. 17. This shows a much clearer relationship between the corrected rainfall field and the rainfall shadows, with strong correlations between the average bias and the average rainfall rate. The relationship becoming less clear for higher rainfall thresholds for conditional average, with the strongest correlation between the non-zero rainfall average and the RMSE.

The ARS in images may increase exponentially with increasing average rainfall rates, however this may be skewed with
the small number of images with very high ARS (larger than $1000\text{km}^2$). For the thresholded rainfall averages of 0, 0.1, 0.5 and 1, up until an average rainfall rate of 0.15, 0.6, 1.5 and $2\text{mmh}^{-1}$, the ARS is consistently low, with low variability. The relationship between the proportion of rainfall in Fig. 17 is more complex, with two distinct types of image behaviour.

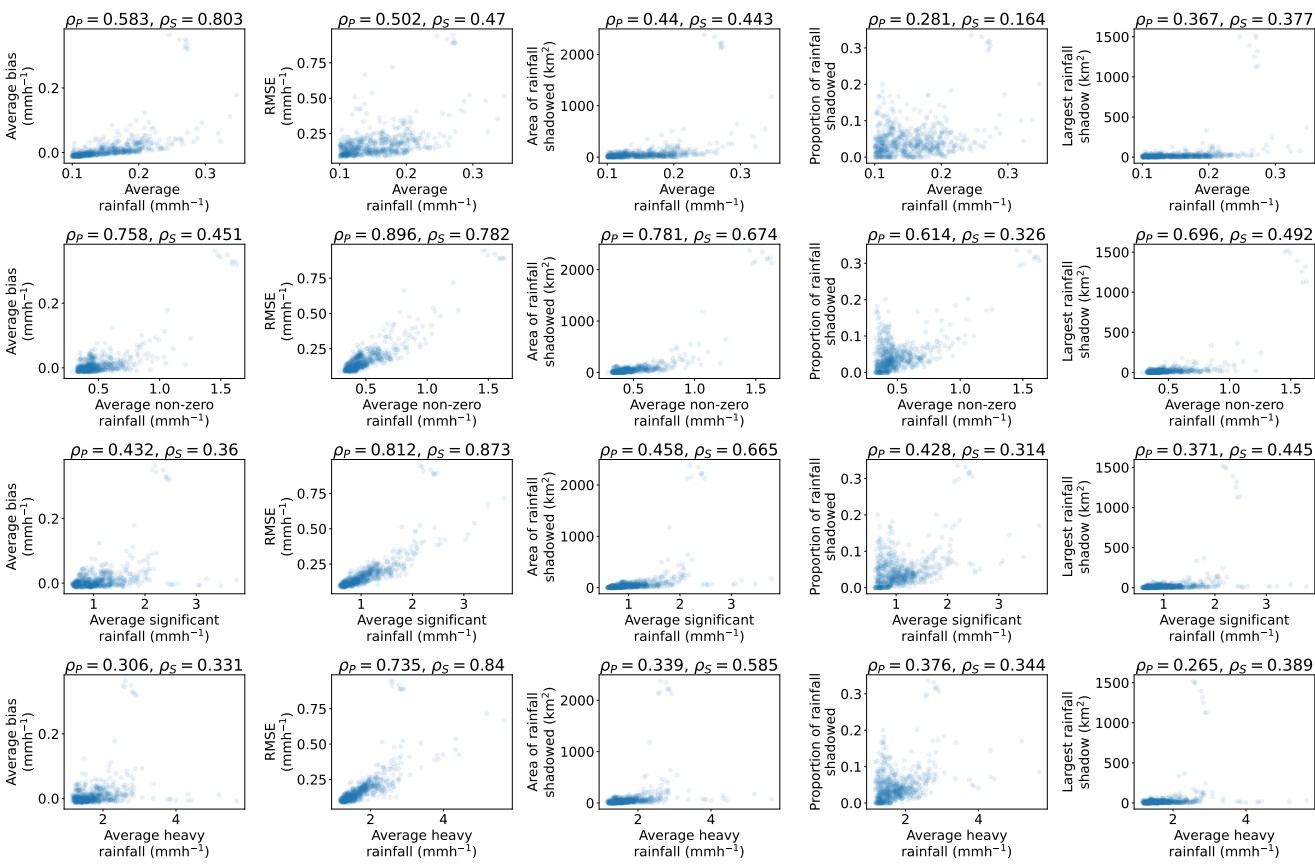

**Figure 17.** Average rainfall (with thresholds 0, 0.1, 0.5 and $1\text{mmh}^{-1}$) and average bias, RMSE, and shadow area, proportion and largest shadow for event images with an average rainfall rate of at least $0.1\text{mmh}^{-1}$




While a correlation is evident between the conditional average rainfall rates and the PRS, there is clearly a large number of images with low average rainfall rates and large PRS. These high PRS with low average rainfall rates are most likely
corresponding to images with a very low proportion of rainfall rates large enough to be classed as shadows. If just one rainfall pixel is shadowed, there would be a large increase in the PRS in this case, highlighting the impact that shadows have in images with low rainfall extents, with fairly low average rainfall rates.

The LARS appear to have a similar relationship to the ARS, however these are again skewed by very high ARS and so the relationship is less clear. Considering the overall average behaviour of the ensemble does not take full advantage of the
model framework, and different ensemble member properties, as important variation information is lost, which may increase understanding of the uncertainty associated with the radar rainfall estimation process.

## 5.2 Ensemble variability

The variability between ensemble members for each event image is considered, using th ensemble standard deviation to identify areas were the event errors have a high level of uncertainty in image properties. Figure 18 shows the relationship between the
average non-zero rainfall rate and the standard deviation of the images bias and RMSE, showing rainfall images with an average non-zero rainfall rate less than 0.5mmh$^{-1}$, there is no clear relationship between the average rainfall and variability in the bias of estimates. For images with an average non-zero rainfall rate above 0.5mmh$^{-1}$, there appears to be a strong positive correlation between the two, suggesting that past this image threshold, the uncertainty in the image bias is directly proportional to the average non-zero rainfall rate.

The relationship between the average non-zero rainfall rate and the image RMSE variability is very different, appearing inversely proportional to the variability in image RMSE. This may be attributed to the fact that for higher rainfall rates, there are more rainfall shadows. In areas where there are rainfall shadows, the variability between ensemble members decreases significantly due to the effects of the radar ray signal being fully damped. There is a moderate relationship between the variability PRS and the average non-zero rainfall, however this is less distinct.

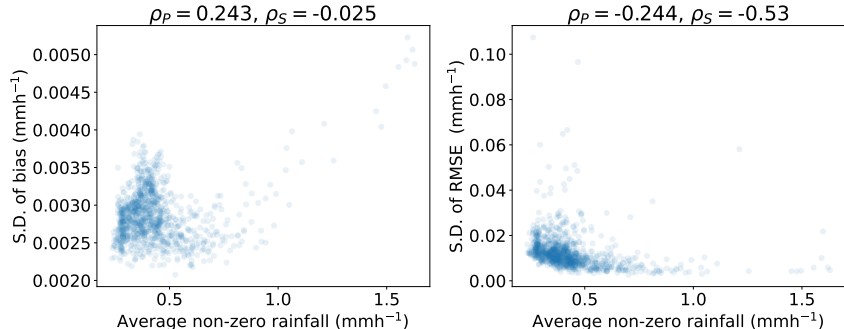

**Figure 18.** Standard deviation of the ensemble bias and RMSE for ensemble images of all events





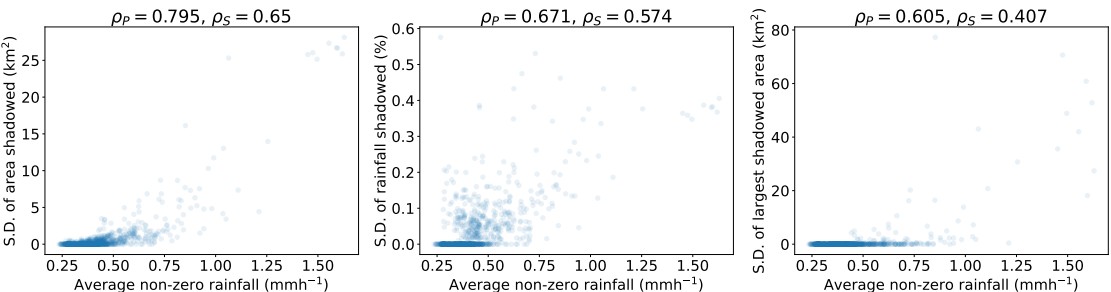

**Figure 19.** Standard deviation of the ensemble area, proportion of rainfall, and largest area shadowed, for event images

The standard deviation of the ARS, PRS, and LARS are given in Fig. 19. From this we can see that the uncertainty in the ARS increases with increasing average rainfall rate, with similar behaviour for the LARS in images, however there are a lot of imageS with no ARS skewing the relationship. Again the relationship between the average non-zero rainfall rate and the variability of the PRS is not clear, as is the case with the average PRS.

## 5.3 Rainfall location : second moment of area

The location of rainfall with respect to the radar location will also impact the error structure, which is reflected in the variability in the PRS for different average rainfall rates. Atlas and Banks (1951) stated that distortion due to range attenuation includes a displacement towards the radar of maximum intensity, packing contours on the near-side of the storm, suggesting that the location of high-intensity rainfall will also have an impact on errors. The amount of rainfall lost due to attenuation effects is likely to be higher for images with high-intensity rainfall in a more central location, due to the cumulative nature of attenuation
effects along a radar ray.

As illustrated in Fig. 20, the closer to radar the rainfall is, the more rays it will impact, and the earlier on in a ray it damps the signal, affecting more radar bins. There is evidence of this in Fig. 7, where high-intensity rainfall occurred in the centre of the image, resulting in very high errors.

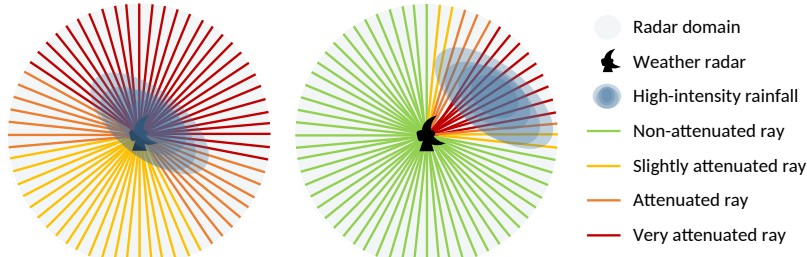

**Figure 20.** Schematic of the different effect high-intensity rainfall has on the radar rays based on its location within the domain





To formally investigate this, we introduce the second moment of area of a rainfall (or reflectivity) image, estimated by considering the centroid of each image using the second areal moment. For rainfall field $R(t)$ on a Cartesian grid with dimensions $(N_x, N_y)$, the second moment of area can be estimated as

$$M_R(t) = \sum_{x=1}^{N_x} \sum_{y=1}^{N_y} R(x,y,t) d(x,y)^2 = \sum_{x=1}^{N_x} \sum_{y=1}^{N_y} R(x,y,t) \left\{ (x-r_x)^2 + (y-r_y)^2 \right\} \tag{11}$$

where $d(x,y) = \sqrt{(x-r_x)^2 + (y-r_y)^2}$ is the distance of a pixel from the radar location $(r_x, r_y)$. Both the actual and normalised rainfall fields are used to see the different impact between the shape of the field, with and without considering the actual magnitude of estimates.

Fig. 21 shows the relationship between the second moment of area and normalised second moment of area for event images, and the corresponding ensemble average bias, RMSE, ARS, LARS and PRS. There are significant differences between the moments and normalised moments, with positive correlations for image moments and an inverse relationship for normalised image moments, which is particularly prominent when considering the RMSE. There is a significant positive correlation between image moment and RMSE, ARS and PRS, with the relationship for the LARS is less clear.

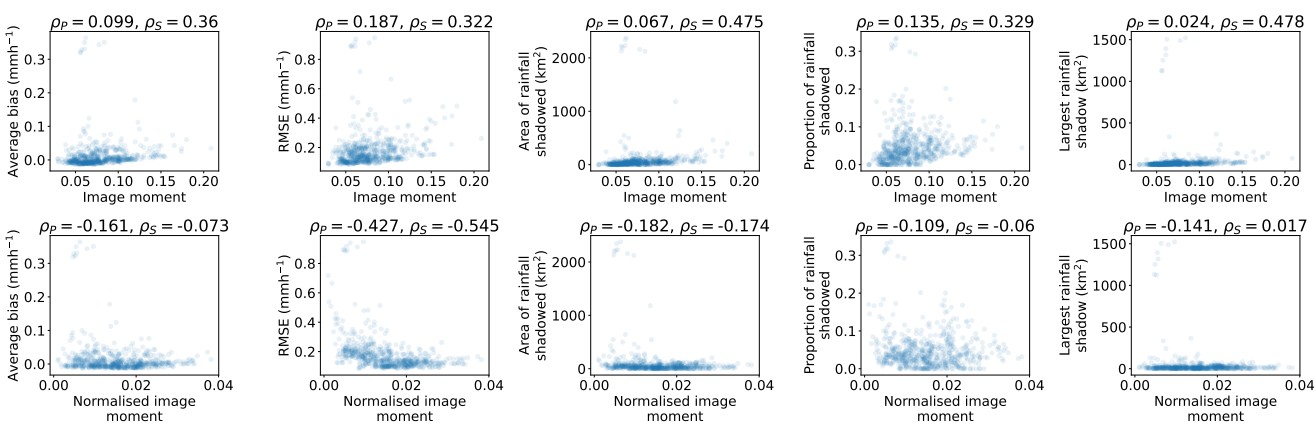

**Figure 21.** (Normalised) second moment of area for the average bias, RMSE, area, proportion and largest area of shadowed rainfall for event images

This suggests that for the second moment of area calculated on rainfall rates, the strong non-linear dependence on absolute rainfall amount is overriding all other information about the field, such as rainfall location. For the normalised second moments of area, this dependence has been removed, and so the relationship is solely based on the impact of the rainfall location. In this case, a smaller second moment of area (corresponding to a more central rainfall location) suggests larger rainfall shadows.

Fig. 22 shows the relationship between the second moment of area and normalised second moment of area for events, and the corresponding ensemble uncertainty in terms of the standard deviation for the bias, ARS, LARS and PRS. This shows that images with a larger image moment have lower variability in the RMSE. The normalised image moments appear to be




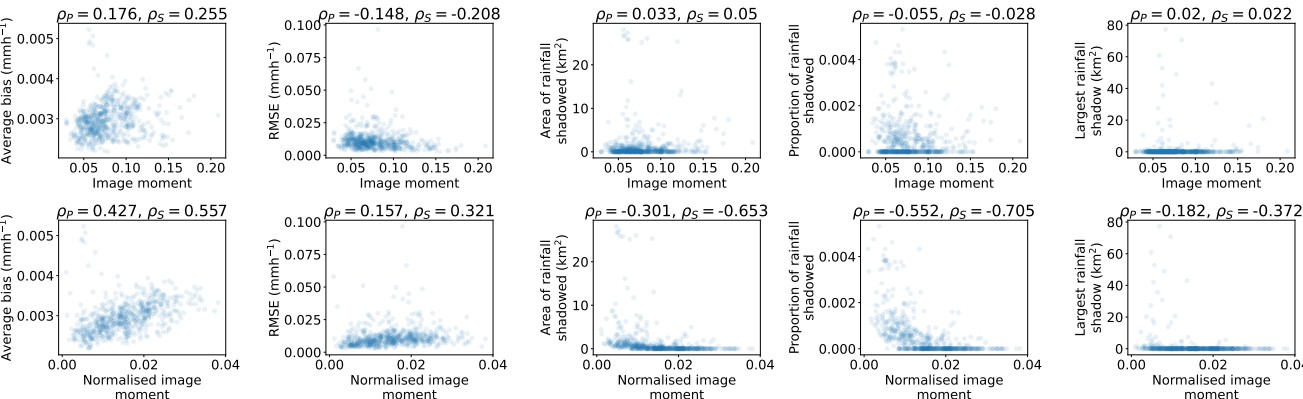

**Figure 22.** (Normalised) second moment of area for the standard deviation of bias, RMSE, area, proportion and largest area of shadowed rainfall for event images

positively correlated with the average bias and RMSE, and negatively correlated with the ARS, PRS and LARS. The variability in rainfall shadows, for both the ARS and PRS, appears to decrease with increasing normalised image moments.

Image moments could be a key piece of information when attempting to identify radar images with high uncertainty in estimates, particularly when using moments calculated from normalised rainfall rates across an image. In conclusion, this analysis suggests that second moment of area has potential in identifying high uncertainty and missing information.

## 5.4 Rainfall shadow frequency

The aim of this section is to identify how often rainfall shadows occur. Due to ensemble variability, to ensure frequencies are not overestimated, we consider the 'best case scenario' over the ensemble. In terms of the ensemble, for each image the ensemble member with the lowest errors is selected. This prevents overestimation, and as rainfall fields are parametrised with existing corrected radar rainfall images that may themselves be subject to rainfall shadows, the simulations may inherently underestimate the frequencies and so considering the minimum likelihood of occurrence makes intuitive sense. Percentiles are given for the minimum LARS, PRS and ARS in Table 1.

| Percentile (%) | Proportion shadowed | Largest shadow | Actual shadow |
|---|---|---|---|
| **25** | 0.01 | 0 | 0 |
| **50** | 0.03 | 10.1 | 15.2 |
| **75** | 0.06 | 20.2 | 45.5 |
| **90** | 0.09 | 30.3 | 90.9 |
| **95** | 0.13 | 50.5 | 136.4 |

**Table 1.** Percentiles for the ECDF for the minimum ensemble rainfall shadow proportions, largest shadowed areas, and actual rainfall shadowed for all events



The empirical cumulative distribution function for the minimum LARS, PRS and ARS are estimated for simulated events, with attenuation estimated from rainfall and reflectivity, given in Fig. 23.

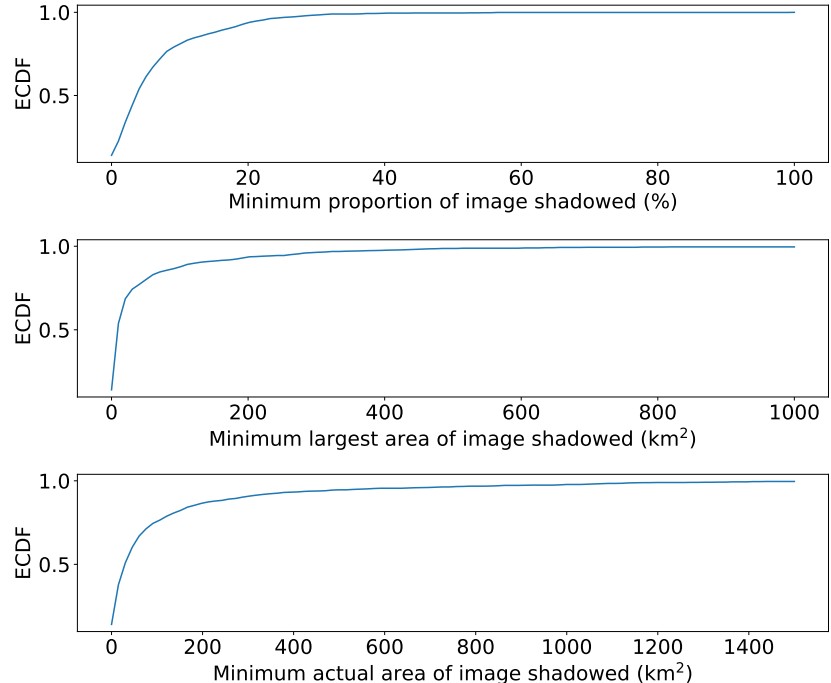

**Figure 23.** Empirical cumulative distribution functions (ECDF) for the minimum ensemble proportion, largest and actual area of image containing rainfall shadows for all events

From the median of proportion of rainfall images simulated, half of the images have at least 3% of significant rainfall rates shadowed. When considering the ARS in images, we see that 25% of images have an ARS of 45km$^2$ and 5% have a LARS of over 50km$^2$. A missing area of significant rainfall of this size, particularly for small or urban catchments, constitutes a major an underestimation of flood risk, resulting in incorrect information provided in flood warnings. This highlights the importance of gaining an improved understanding of rainfall shadows, and provides motivation for this project and future research in this area. Gaps caused by the rainfall shadows identified would result in under-prediction of flooding, impacting both flood warnings and flood defence designs.

## 6 Discussion and conclusions

Errors relating to several different aspects of the radar rainfall estimation process are considered, using a radar error model outlined in detail. This model is applied to realistic simulated rainfall events in a stochastic manner, generating an ensemble of radar images corresponding to each time step of rainfall events. A Log-Normal random noise field was imposed on rainfall



estimates to account for underlying non-specific noise. The DSD uncertainty is included by replacing the multiplicative parameter $a$ in the $ZR$-relationship with a two-dimensional spectral random field, with field variability determined by radar sampling volumes. Attenuation effects are imposed by inverting standard gate-by-gate correction algorithms (Jacobi and Heistermann, 2016). To enable the direct comparison between the simulated rainfall before and after imposing the radar error model, each

radar image is corrected using a standard radar rainfall estimation process. This results in corrected rainfall field for each ensemble member, similar to what would be obtained from real radar rainfall images, allowing us to identify when and where significant and/or systematic errors may occur.

This concept provides a methodology for developing a better understanding of errors relating the the radar rainfall estimation process. By generating a 'true' rainfall field, and subsequently imposing errors to allow for the comparison with correct 'best

guess' stochastic radar rainfall estimates, allows us to address the fundamental limitation of weather radar correction schemes – that the real rainfall field is not known for comparisons. An investigation of the spatio-temporal behaviour of the error structure is then possible, which provides key information about the radar rainfall estimation process.

A relationship between rainfall shadows, high bias and uncertainty, related to the amount of rainfall (i.e. proportion and rates) in images was found. The impact rainfall location with respect to the weather radar is considered by introducing the second

moment of area, showing more central rainfall in the radar domain results in higher errors and variability. The minimum likelihood of rainfall shadows showed that 50% of images simulated have at least 3% of significant rainfall shadowed. In addition, 25% of images had an ARS of over 45km$^2$, with the minimum LARS for 5% of images exceeding an area of 50km$^2$. This gap would result in underestimation of potential impacts of flooding. This highlights the importance of gaining an improved understanding of rainfall shadows, and provides motivation for this project, and future research in this area. Weather

radar alone cannot be used for rainfall estimation, as information is regularly missed.

## 6.1    Impact and transferability

Improved high-resolution rainfall estimates are needed for flood forecasting, by stakeholders and water managers, particularly in (near) real time, for nowcasting and probable ensemble forecasting. The radar error model outlined in this study produces visually realistic radar images, capturing key properties of radar images, with many potential uses for the model framework,

some of which are outlined below.

1. Identification of radar rainfall properties which correspond to high errors and uncertainties could be extended and used in a probabilistic manner, to better condition merged rainfall fields. Identifying areas where the spatial distribution of the rainfall cannot be trusted (i.e. occasions where rainfall shadows are likely, such as areas past high intensity rainfall). This includes information on the frequency and location of rainfall shadows as a merging criterion (i.e. putting more weight

on rainfall information from other sources when rainfall shadows are likely to occur at a given location).

2. Application of the model to gridded rain gauge fields or forecasts, for comparison with corresponding weather radar images, to better identify and understand radar rainfall errors.



3. The importance of complex and efficient radar gauge merging methods is emphasised in this study, which do not trust the spatial distribution of rainfall provided by weather radars alone. Additional information from other sensors is needed, such as opportunistic sensors, citizen science data, rain gauges and microwave links. This study has provided a framework for methods of assessing the performance of these merging techniques.

4. Determining the optimal location for weather radars or rain gauges, such as establishing rain gauge networks in areas where rainfall shadows are more likely, and not in densely populated areas (e.g. the city), where it may be too late for warnings of missing information.

5. Assess how radar rainfall errors propagate into hydrological and hydrodynamic modelling, considering the impact that the incorrect distribution of rainfall has on discharge and flood depths.

## 6.2 Limitations and future work

A powerful framework for investigating radar rainfall errors has been developed and demonstrated, with model design allowing for a high degree of flexibility and several natural extensions. The influence of different correction methods for radar rainfall estimation process and the impact this has on the error structure should be investigated using the methodology in this study.

Some error sources in radar rainfall estimation are not included in the radar error model, as they were beyond the scope of this study. To improve the model, additional sources of error could be easily included (e.g. radar calibration errors using an additive error in DBZ and bright band effects using a vertical representation for different weather types and seasons of events). Mountainous regions are typically subject to more errors due to beam blockage from topography, however this could be included in the model quite easily by an additive error based on existing clutter maps from the weather radar of interest. It would be interesting to repeat the study with different DSD structures, changing the correlation structure and marginal distribution of this (previously Gaussian) field. A dependence between DSD parameters could be imposed, as a varying $ZR$-relationship in space and time improved rainfall accumulations at event scale (Libertino et al., 2015).

Close to the radar, measurement volumes are small, systematically increasing in size with distance from the radar. Although as part of the radar error model, the spatial sampling aspect is considered through the estimation variance, radar measurements are taken as instantaneously (as opposed to rain gauge measurements which are temporal aggregations by definition). Implications of these space-time sampling properties mean that temporally we only have a snapshot of the pixel behaviour at a given time. Correlation structure variability in space and time was incorporated through a spatio-temporal anisotropy factor, without explicitly account for the different data sampling between the two dimensions. The simulation environment could be modified to account for the temporal sampling issue by simulating temporally at a higher resolution than existing radar images, and sampling these to reproduce the snapshot effect. Methods could also be developed within an inverse modelling framework (Grundmann et al., 2019) for obtaining field uncertainty in (near) real time.





### 6.3 Concluding remarks

The overarching aim of this study is to contribute towards improvements in the radar rainfall estimation process, by gaining an improved understanding of the frequency and location of the error structure relating to the process. With this in mind, we explore and exploit space-time properties of rainfall and reflectivity, to gain an improved understanding of the error structure between the two, investigating the extent of uncertainties in the radar rainfall estimation process. This study has presented an innovative model for investigating uncertainties in the radar rainfall estimation process; providing a flexible tool that has many potential future applications. The radar error model, outlined in detail, generates a stochastic ensemble of radar images corresponding to an existing rainfall field, by inverting the radar rainfall estimation process. This model incorporates many different error sources, including: the drop-size distribution, attenuation effects, random noise and radar sampling. This provides a method for identifying when and where radar errors are likely to occur, and how often information about the rainfall field is lost, significantly impacting the spatial rainfall field. The insights from this study provide an improved understanding of the error structure between rainfall and reflectivity, and the extent of uncertainties in the radar estimation process. A framework has been provided for investigating the impact of errors relating to the radar rainfall estimation process, with many potential hydrological applications.

*Author contributions.* Amy Green: Conceptualisation , Methodology, Software, Formal analysis, Investigation, Data Curation, Writing - Original Draft, Visualization. Chris Kilsby: Conceptualisation, Methodology, Writing - Review & Editing, Supervision. András Bárdossy: Conceptualisation, Methodology, Supervision.

*Competing interests.* Author Andras Bardossy is an editor for HESS.

*Acknowledgements.* This work is funded by the Natural Environment Research Council (NERC) sponsored Data, Risk and Environmental Analytical Methods (DREAM) Centre for Doctoral Training in risk and mitigation research using big data (NE/M009009/1).



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
