# Peer review of "Assessing rainfall radar errors with an inverse stochastic modelling framework"

_EGUsphere, 2024_

## Referee Comment (RC2)

**Review of *Assessing rainfall radar errors with an inverse stochastic modelling framework* by Green et al. (2024)**

March 25, 2024

**1  Introduction**

The presented work makes use of the rainfall simulator presented in Green et al. (2023). Using the generated rainfall fields as ground truth it works backwards taking into account multiple error sources to generate the equivalent radar reflectivity that would be observed by a C-band radar. From this radar reflectivity another corrected 'best guess' rainfall field is obtained. The ground truth is then compared with this best-guess and the results are used to assess the uncertainties in radar QPE. Validating a QPE method is challenging as the ground truth is typically only available very locally at rain gauges or integrated in time over watersheds. Therefore, this work proposes an interesting approach which could be promising. Unfortunately I found the methodology to be difficult to understand as some key explanations are missing (in particular how the best-guess rainfall field is estimated and how beam-broadening/radar-sampling is taken into account) and the work could benefit from an in-depth review of a weather radar expert (preferably from the MetOffice), as I feel that it could be really beneficial to the quality of the work. In the end, I think that the work is worth publishing but the following points would need to be addressed.

**2  Major points**

1. l140: Unfortunately several lines are missing which makes it very difficult to properly understand this paragraph. Moreover I find that some clarifications and derivations are missing (in particular for Equ9). This makes this subsection really difficult to follow, which is unfortunate given how important it is in the methodology. I would suggest to rewrite this section entirely. Figure 4 also needs significant improvements as its relation with Eq 8 and 9 is not clear at all. A height discretization is briefly mentioned without any details. This discretization should be thoroughly explained and described as well in Figure 4.

2. I'm not sure how the corrected rainfall field (last plot in Fig 6 for example) is obtained and it is not clear in the text. There is a missing explanation on how to go from the ensemble of attenuated reflectivities (last panel of Fig 5.) to this corrected rainfall field. Is it the same QPE used in Green et al. (2023)? Even in your rainfall simulator paper, the QPE method is not described in detail. In the conclusion it is written *a standard radar rainfall estimation process*, however that does not mean much to me as QPE methods are typically quite specific and are often fine-tuned by the weather service that operates them. Additional explanations are needed in particular as it is the main output of the whole method!

3. Sec 3.3: you have some very low values of reflectivity in the histogram of reflectivity ($< -20$dBZ). I suspect that you considered raw radar data, without any significant noise filtering to derive this statistic. However, in practice every standard weather radar processing routine will remove these values by applying a minimum SNR. As such, I don't feel like your approach is representative of a real use case. I would suggest to use the same type of noise thresholding used at the MetOffice on your radar data and fit a model only to the remaining unfiltered noise. I suspect that in the current situation the resulting binary precipitation field (precip/no-precip) will look very unrealistic due to not filtering any noise (see next point).

4. Choice of colormaps: the choice of a default colormap for all variables (viridis) is not really appropriate. Typically precipitation intensity and reflectivity are displayed with variations of the Rainbow colormap, which makes it easier to detect regions of large precipitation. Other variables which are bounded (for example proportional error) should rather be shown with a sequential colormap. I would suggest to check the pyart library for relevant colormaps for reflectivity and precipitation. Another important thing is that you should separate no-precip (dry) from low precip to improve visibility. The values of zero precip should be left white in my opinion.

5. Equ 11: Fig. 20 shows that the effect of attenuation is maximal if $R$ is large and the distance $d$ to the radar is small, but $MR(t)$ is largest if the distance is large and $R$ large. As such, why is it used as an indicator for the effect shown in Fig 20. Shouldn't we choose an indicator that is inversely proportional to $d$ ?

6. Fig16 and 17: these figures show empty plots, without any explanation why!

7. Fig21 and 22: : I don't get the difference between the two figures. It is not clear at all in the text. Is the second image for precipitation $> 0.1$ mmh$^{-1}$ only?

**3  Minor points**

1. l.21:I think you should also include the beam-broadening effect with non-uniform beam filling in the major error sources for QPE

2. l.52: Missing reference, should be Section 3

3. l.61: Since the study focuses on a single radar, it would be good to provide some info on this radar for people who are not familiar with the Met Office radar network (e.g. coordinates, frequency, PRF, pulse width)

4. Equ3: please introduce the precipitation intensity variable $R(t)$ properly

5. Fig3: please indicate clearly in the caption that the blue histogram is empirical from radar data.

6. L133 eq8: $c$ is not defined

7. l.160: what is pixel variability? I imagine it is the variability between ensembles at pixel level. But this is defined and explained nowhere. The proportional error is also not defined explicitly.

8. Fig5: Please provide proper coordinates for your radar image, both in X and Y directions (-100 to 100 km). Do it at least in one image, you can then potentially indicate that, since the coordinates are always the same, you drop them from your figures. I saw that on Fig 7. 10 and 13, you indicated the coordinates, but starting from the top-left of your domain which is not standard. Please make everything more homogeneous.

9. Fig12: the area of high precipitation is not visible in the image because the color scale of the rainfall field is not adequate. If the area is too small, use at least an arrow or a circle to show it. Also I wonder to which extent a precipitation field with such a tiny region of extremely high precipitation surrounded by almost zero everywhere else is realistic in a real world case (and is not caused by unfiltered clutter).

10. l.228: I don't understand *proportion of rainfall in an images with the image RMSE*, shouldn't it be proportion of heavy rainfall as in the plot y-label. And if yes, how is "heavy rainfall" defined? Is it the same as the significant rainfall from the PRS (1mmh$^{-1}$), because in this case heavy is maybe a bit exaggerated?

11. l.277: "imageS" (wrong capital S)

12. Tab1: Please provide full name of ECDF here (empirical cumulative distribution function) as it the acronym is used before Fig23.

13. The addition of letters (a, b, c, . . . ) in the captions of multiple panel figures (e.g. Fig 11, 22,...) would make it much easier for the reader to immediately link the corresponding plot to the reference in the caption.

---

## Author Comment (AC2)

**RC2**: 'Comment on egusphere-2024-26', Anonymous Referee #2, 25 Mar 2024 reply

*The presented work makes use of the rainfall simulator presented in Green et al. (2023). Using the generated rainfall fields as ground truth it works backwards taking into account multiple error sources to generate the equivalent radar reflectivity that would be observed by a C-band radar. From this radar reflectivity another corrected 'best guess' rainfall field is obtained. The ground truth is then compared with this best-guess and the results are used to assess the uncertainties in radar QPE. Validating a QPE method is challenging as the ground truth is typically only available very locally at rain gauges or integrated in time over watersheds. Therefore, this work proposes an interesting approach which could be promising. Unfortunately I found the methodology to be difficult to understand as some key explanations are missing (in particular how the best-guess rainfall field is estimated and how beam broadening/radar-sampling is taken into account) and the work could benefit from an in-depth review of a weather radar expert (preferably from the MetOffice), as I feel that it could be really beneficial to the quality of the work. In the end, I think that the work is worth publishing but the following points would need to be addressed.*

The authors would like to thank reviewer 2 for the in depth comments, and appreciate the time and effort made and insightful comments for improvements to our paper. We thank the reviewer for pointing out the need for clarification on methodology and agree that key explanations (e.g. how the best guess rainfall field is estimated and how beam broadening/radar-sampling is taken into account) should be expanded upon and clarified and will be done in the revised manuscript. Below is a breakdown of author comments in response to the detailed major and minor points given by reviewer 2.

**Major points to address:**

1. *l140: Unfortunately several lines are missing which makes it very difficult to properly understand this paragraph. Moreover I find that some clarifications and derivations are missing (in particular for Equ9). This makes this subsection really difficult to follow, which is unfortunate given how important it is in the methodology. I would suggest to rewrite this section entirely. Figure 4 also needs significant improvements as its relation with Eq 8 and 9 is not clear at all. A height discretization is briefly mentioned without any details. This discretization should be thoroughly explained and described as well in Figure 4.*

The authors would like to thank reviewer 2 for noticing the missing lines, and would like to apologise for this formatting issue with the template that we missed. This will be included in the revised manuscript. The authors appreciate that Section 3.5 is difficult to follow, and propose significant changes in the revised manuscript. Updates to Figure 4 have been made (see below) to include the discretisation and sampling volumes which link better to Equation 8 and Equation 9. Additional explanation will be given on how random sampling will be accounted for in the model, with a brief summary given below, and more detailed figure caption.
  - For a given radar pixel, we consider a vertical column of rainfall, assuming that the cloud base, bright band and cloud top are 1, 4 and 10km. For the vertical column of rainfall, we

assume a DSD with an exponential variogram based on vertical radar measurements and Berne et al (2005), for below the bright band level.

- Discretising the vertical column of rainfall into blocks of size 10m, the empirical distances for each discretised block are calculated, and the variogram model is used to give gamma values for each distance, and the average is taken to get the sampling variance for the whole column $V$.
- Next we consider the sampled blocks the radar configuration scan angles, for each grid point (see Fig 4. highlighted yellow). For these visible blocks we again calculate the distances and corresponding gamma values, to give the sampling variance for each $v_i$.
- The estimation variance is then the sampling variance (or mean variogram) for visible blocks, subtracted from that of the whole column.

[Figure]

2. *I'm not sure how the corrected rainfall field (last plot in Fig 6 for example) is obtained and it is not clear in the text. There is a missing explanation on how to go from the ensemble of attenuated reflectivities (last panel of Fig 5.) to this corrected rainfall field. Is it the same QPE used in Green et al. (2023)? Even in your rainfall simulator paper, the QPE method is not described in detail. In the conclusion it is written a standard radar rainfall estimation process, however that does not mean much to me as QPE methods are typically quite specific and are often fine-tuned by the weather service that operates them. Additional explanations are needed in particular as it is the main output of the whole method!*

The authors would like to thank the reviewers for pointing out this missing information, which was mostly provided in the missing lines (see below), and would again like to apologise for this formatting issue with the template that we missed. This will be included in the revised manuscript, with additional details on the specific QPE used in this case. This QPE followed C-band single-polarisation, using the *wradlib* package on Python. This includes a PIA attenuation correction (Kraemer & Verworn, 2008), ZR-relationship to obtain rainfall rates and reprojection to a Cartesian grid.

**4 Results: specific events**

*In this section, the performance of the model is considered by looking at example fields for each step of the simulation process. To enable direct comparisons, each radar image is corrected using a typical radar rainfall estimation process, resulting in a corrected rainfall field for each ensemble member, similar to what would be obtained from a radar*

*rainfall images. Defining the difference between the simulated `true' rainfall field and the rainfall field after applying the radar error model as the error, we consider three different events. To investigate the capacity of the radar error model to capture uncertainty across a rainfall event, we consider the overall behaviour and variability of pixels throughout the stochastic ensemble of radar images. For a rainfall event or image denoted by $\hat{R}$, the bias and root-mean square error (RMSE) are considered   < RMSE and BIAS equations >  for corrected rainfall field $R_{corr,i}$ of ensemble member $i$, where $i = 1, ..., n$ for $n = 100$. The pixel variability, proportional error and RMSE limits (minimum and maximum over ensemble) are also calculated, and the proportional error taken as the ratio between the simulated and corrected rainfall field for all non-zero simulated rainfall pixels. For illustrative purposes a single time step towards the middle of the event is included, with a link to event videos included in each figure.*

3. *Sec 3.3: you have some very low values of reflectivity in the histogram of reflectivity (< -20dBZ). I suspect that you considered raw radar data, without any significant noise filtering to derive this statistic. However, in practice every standard weather radar processing routine will remove these values by applying a minimum SNR. As such, I don't feel like your approach is representative of a real use case. I would suggest to use the same type of noise thresholding used at the MetOffice on your radar data and fit a model only to the remaining unfiltered noise. I suspect that in the current situation the resulting binary precipitation field (precip/no-precip) will look very unrealistic due to not filtering any noise (see next point).*

In this study, we used the raw dBZ values in the form provided by the UK Met Office, and followed the dual-polarisation radar rainfall estimation workflow as outlined on *wradlib*, which we take to be a source of widely used standard practice. For the methodology we have designed, it is actually important not to filter out additional noise, as it could cumulatively impact rainfall estimates along the radar ray, and additionally rainfall rates of 0 would create instability when convert from rainfall to dBZ. In terms of comparison with "standard" Met Office methodology, The processing methods obtained similar results from processing as Met Office rainfall estimates (which correspond to rainfall rates from -32dbZ and above). Including the random noise in the model is a crucial element, and does not impact the rainfall /no rainfall distribution as this is thresholded later. From Figure 2 and Figure 3 we can see that the non-zero rainfall threshold (blue line) is higher than the random noise distribution, and an example dry day image. Additional clarification can be added to the revised manuscript to clarify this if necessary.

4. *Choice of colormaps: the choice of a default colormap for all variables (viridis) is not really appropriate. Typically precipitation intensity and reflectivity are displayed with variations of the Rainbow colormap, which makes it easier to detect regions of large precipitation. Other variables which are bounded (for example proportional error) should rather be shown with a sequential colormap. I would suggest to check the pyart library for relevant colormaps for reflectivity and precipitation. Another important thing is that you should separate no-precip (dry) from low precip to improve visibility. The values of zero precip should be left white in my opinion.*

While the authors agree that rainbow colour maps have been commonly used for precipitation diagrams, numerous recent studies have recommended against their use. General and hydrology specific comments on selecting appropriate colour maps are listed below. Additionally, we have consulted with a geospatial imagery specialist on the subject. As a result of extensively investigating and discussing the issue, we propose to follow the specialist advice recommending against rainbow colour maps.

https://ica-abs.copernicus.org/articles/1/96/2019/
https://www.sciencedirect.com/science/article/pii/S0924271622002659
https://hess.copernicus.org/articles/25/4549/2021/

An example of the reasoning is: "The disadvantages most often raised are as follows: (1) it is not perceptually ordered and there is no logical ordering (Monmonier 1991), which means there is no innate sense of higher or lower values; (2) it introduces sharp transitions between hues as the perceptual changes in the rainbow colours are not uniform (Moreland 2016), and which may be perceived as being a considerable transition in the mapped data; and (3) the middle values may be interpreted as extreme values since yellow has a highlighting effect, being perceived as brighter than the other colours." - https://doi.org/10.5194/ica-abs-1-96-2019

A more monotonic and appropriate rainbow colour map could be selected; however this introduces additional issues (e.g. similar colours but different colour scale to conventional rainfall rainbow colour maps).

In addition, the authors consider accessibility to be very important, and the selection of a rainbow colour has inclusivity issues, due to the fact that they are not colourblind safe. The authors are happy to consider other colour maps that are colour blind safe, and do not have mid-range values as white as this can also be difficult to understand, but on balance we and other readers are happy with the existing scheme.

5. *Equ 11: Fig. 20 shows that the effect of attenuation is maximal if R is large and the distance d to the radar is small, but MR(t) is largest if the distance is large and R large. As such, why is it used as an indicator for the effect shown in Fig 20. Shouldn't we choose an indicator that is inversely proportional to d ?*

While we agree that there may be an advantage in an indicator which co-varies with attenuation directly, and could adapt the measure to be an inverse moment measure, we have found that this then puts a much greater emphasis on extremely large values, which is a very small subset of events. We feel that the existing measure is still intuitive to understand and based on the simplest available physical principles, and so prefer to leave as it is.

6. *Fig16 and 17: these figures show empty plots, without any explanation why!*

The authors are confused as to which plots are empty, as online version we have access to has visible plots for Figure 16 and Figure 17. However, if there is an issue the authors have attached copies of these plots for clarification, and if there are any additional issues please let us know.

7. *Fig21 and 22: : I don't get the difference between the two figures. It is not clear at all in the text. Is the second image for precipitation > 0.1 mmh−1 only?*

We agree, thankyou for raising this. From the captions it is not clear the difference between Figure 21 and Figure 22 and unclear in the text. Figure 21 shows the average ensemble behaviour, whilst Figure 22 show the ensemble variability (i.e. how much the RMSE or bias varies throughout the ensemble). The authors will clarify this in the revised manuscript, in both the captions and corresponding text, with a more concise Section 5.3 in general.

**Minor points to address:**

1. *l.21:I think you should also include the beam-broadening effect with non-uniform beam filling in the major error sources for QPE*
2. *l.52: Missing reference, should be Section 3*
3. *l.61: Since the study focuses on a single radar, it would be good to provide some info on this radar for people who are not familiar with the Met Office radar network (e.g. coordinates, frequency, PRF, pulse width)*
4. *Equ3: please introduce the precipitation intensity variable R(t) properly*
5. *Fig3: please indicate clearly in the caption that the blue histogram is empirical from radar data.*
6. *L133 eq8: c is not defined*
7. *l.160: what is pixel variability? I imagine it is the variability between ensembles at pixel level. But this is defined and explained nowhere. The proportional error is also not defined explicitly.*
8. *Fig5: Please provide proper coordinates for your radar image, both in X and Y directions (-100 to 100 km). Do it at least in one image, you can then potentially indicate that, since the coordinates are always the same, you drop them from your figures. I saw that on Fig 7. 10 and 13, you indicated the coordinates, but starting from the top-left of your domain which is not standard. Please make everything more homogeneous.*
9. *Fig12: the area of high precipitation is not visible in the image because the color scale of the rainfall field is not adequate. If the area is too small, use at least an arrow or a circle to show it. Also I wonder to which extent a precipitation field with such a tiny region of extremely high precipitation surrounded by almost zero everywhere else is realistic in a real world case (and is not caused by unfiltered clutter).*
10. *l.228: I don't understand proportion of rainfall in an images with the image RMSE, shouldn't it be proportion of heavy rainfall as in the plot y-label. And if yes, how is "heavy rainfall" defined? Is it the same as the significant rainfall from the PRS (1mmh−1 ), because in this case heavy is maybe a bit exaggerated?*
11. *l.277: "imageS" (wrong capital S)*

12. *Tab1: Please provide full name of ECDF here (empirical cumulative distribution function) as it the acronym is used before Fig23.*
13. *The addition of letters (a, b, c, . . . ) in the captions of multiple panel figures (e.g. Fig 11, 22,...) would make it much easier for the reader to immediately link the corresponding plot to the reference in the caption.*

The authors agree with all comments in the minor points and would like to thank the reviewers for these suggestions. They will be included and addressed in the revised manuscript.

Regarding the realistic behaviour of a small area of high-intensity precipitation in point 9, the authors agree that this is less visible and should be highlighted in a revised manuscript. We believe that this rainfall field is however realistic. Results from this manuscript highlight that areas of precipitation like this may often be missed in radar data due to rainfall shadows, and small areas of high-intensity rainfall are more likely to be missed by event a dense rain gauge network.

---

## Author Response (AR1)

**Reviewer comments**

RC1: 'Comment on egusphere-2024-26', Anonymous Referee #1, 25 Mar 2024 reply

The authors discuss weather radar measurements and attempt to quantify the frequency and extent to which important rainfall information is captured in these measurements. Clearly, such measurements are paramount for detailed recording of storms and precipitation. However, error sources like attenuation and ground clutter can obscure these rainfall measurements. The authors introduce a novel approach by using a stochastic model to simulate rainfall data, which is considered accurate. They then systematically introduce errors to this data to emulate the radar estimation process and study the error patterns. The model effectively generates these images for various event types, aiding in understanding and correcting radar rainfall estimation errors.

Honestly, there is not much to write about the paper. The paper is clear, well-structured, and crafted, supported by informative and high-quality figures and equations. Of course, there are several assumptions used but overall, the methods are robust. The examples provided for high bias, low and high variability are informative and indicative, alongside the individual image-based results. The work also acknowledges potential limitations and suggests possible extensions. I suggest the authors have a careful look for typos, e.g.: line 41 ... is that..., line 53 ... Section ??. Also try to provide some additional information about the methodological choices that are just mentioned, e.g., line 91. ...a log-normal... indeed simple but is it the right distribution for this case? Or line 92 - how did you estimate the anisotropy? Concluding, I believe this is a high-quality technical study that achieves its intended goal and adds to the literature.

**Response**

The authors thank reviewer 1 for the positive comments, and appreciate the time and effort made and insightful comments for improvements to our paper. We thank the reviewer for pointing out two typos in the manuscript (line 41 and line 53), which are corrected in the revised manuscript.

We agree with the comments made about clarifying methodological assumptions on line 91 (log-normal assumptions) and line 92 (anisotropy), and have added the following clarification to the revised manuscript in Section 3.3:

*"The random noise field is added to rainfall values to prevent numerical instabilities, with the marginal distribution from Fig. 2 converted to rainfall rates in Fig. 3. When considering the logarithm of weather radar noise (i.e. dry day images and values of dBZ corresponding to rainfall rates less than 0.1mmh$^{-1}$), these are sufficiently Gaussian to satisfy the assumption of a Log-Normal marginal distribution for random noise effects. A Log-Normal marginal distribution allows for a simple and easy transformation when simulating the field using Gaussian random field theory. Empirical variograms of these*

*values were estimated to identify an appropriate correlation structure, which has a very short correlation range of around 5km. The optimal spatial transformation for minimising least squares between the marginal variogram values of the two spatial dimensions is used to estimate field anisotropy from empirical variogram fields, with estimates suggesting that isotropy of random noise fields is a valid assumption in this case."*

**RC2**: 'Comment on egusphere-2024-26', Anonymous Referee #2, 25 Mar 2024 reply

*The presented work makes use of the rainfall simulator presented in Green et al. (2023). Using the generated rainfall fields as ground truth it works backwards taking into account multiple error sources to generate the equivalent radar reflectivity that would be observed by a C-band radar. From this radar reflectivity another corrected 'best guess' rainfall field is obtained. The ground truth is then compared with this best-guess and the results are used to assess the uncertainties in radar QPE. Validating a QPE method is challenging as the ground truth is typically only available very locally at rain gauges or integrated in time over watersheds. Therefore, this work proposes an interesting approach which could be promising. Unfortunately I found the methodology to be difficult to understand as some key explanations are missing (in particular how the best-guess rainfall field is estimated and how beam broadening/radar-sampling is taken into account) and the work could benefit from an in-depth review of a weather radar expert (preferably from the MetOffice), as I feel that it could be really beneficial to the quality of the work. In the end, I think that the work is worth publishing but the following points would need to be addressed.*

The authors would like to thank reviewer 2 for the in depth comments, and appreciate the time and effort made and insightful comments for improvements to our paper. We thank the reviewer for pointing out the need for clarification on methodology and agree that key explanations (e.g. how the best guess rainfall field is estimated and how beam broadening/radar-sampling is taken into account) should be expanded upon and clarified and will be done in the revised manuscript. Below is a breakdown of author comments in response to the detailed major and minor points given by reviewer 2.

**Major points to address:**

1. *l140: Unfortunately several lines are missing which makes it very difficult to properly understand this paragraph. Moreover I find that some clarifications and derivations are missing (in particular for Equ9). This makes this subsection really difficult to follow, which is unfortunate given how important it is in the methodology. I would suggest to rewrite this section entirely. Figure 4 also needs significant improvements as its relation with Eq 8 and 9 is not clear at all. A height discretization is briefly mentioned without any details. This discretization should be thoroughly explained and described as well in Figure 4.*

The authors would like to thank reviewer 2 for noticing the missing lines, and would like to apologise for this formatting issue with the template that we missed. This will be included in the revised manuscript. The authors appreciate that Section 3.5 is difficult to follow, and propose significant changes in the revised manuscript. Updates to Figure 4 have been made, with additional figures also added to include the discretisation and sampling volumes which link better to the text. Section 3.5 has been rewritten and more appropriate equations have been selected, with additional explanation will be given on how random sampling will be accounted for in the model, and more detailed figure caption.

2. *I'm not sure how the corrected rainfall field (last plot in Fig 6 for example) is obtained and it is not clear in the text. There is a missing explanation on how to go from the ensemble of attenuated reflectivities (last panel of Fig 5.) to this corrected rainfall field. Is it the same QPE used in Green et al. (2023)? Even in your rainfall simulator paper, the QPE method is not described in detail. In the conclusion it is written a standard radar rainfall estimation process, however that does not mean much to me as QPE methods are typically quite specific and are often fine-tuned by the weather service that operates them. Additional explanations are needed in particular as it is the main output of the whole method!*

The authors would like to thank the reviewers for pointing out this missing information, which was mostly provided in the missing lines, and would again like to apologise for this formatting issue with the template that we missed. This will is included in the revised manuscript, with additional details on the specific QPE used in this case.

3. *Sec 3.3: you have some very low values of reflectivity in the histogram of reflectivity (< -20dBZ). I suspect that you considered raw radar data, without any significant noise filtering to derive this statistic. However, in practice every standard weather radar processing routine will remove these values by applying a minimum SNR. As such, I don't feel like your approach is representative of a real use case. I would suggest to use the same type of noise thresholding used at the MetOffice on your radar data and fit a model only to the remaining unfiltered noise. I suspect that in the current situation the resulting binary precipitation field (precip/no-precip) will look very unrealistic due to not filtering any noise (see next point).*

In this study, we used the raw dBZ values in the form provided by the UK Met Office, and followed the dual-polarisation radar rainfall estimation workflow as outlined on *wradlib*, which we take to be a source of widely used standard practice. For the methodology we have designed, it is actually important not to filter out additional noise, as it could cumulatively impact rainfall estimates along the radar ray, and additionally rainfall rates of 0 would create instability when convert from rainfall to dBZ. In terms of comparison with "standard" Met Office methodology, The processing methods obtained similar results from processing as Met Office rainfall estimates (which correspond to rainfall rates from -32dbZ and above). Including the random noise in the model is a crucial element, and does not impact the rainfall /no rainfall distribution as this is thresholded later. From Figure 2 and Figure 3 we can see that the non-zero rainfall threshold (blue line) is higher than the random noise distribution, and an example dry day image. Additional clarification can be added to the revised manuscript to clarify this if necessary.

4. *Choice of colormaps: the choice of a default colormap for all variables (viridis) is not really appropriate. Typically precipitation intensity and reflectivity are displayed with variations of the Rainbow colormap, which makes it easier to detect regions of large precipitation. Other variables which are bounded (for example proportional error) should rather be shown with a sequential colormap. I would suggest to check the pyart library for relevant colormaps for reflectivity and precipitation. Another important thing is that you should separate no-precip (dry) from low precip to improve visibility. The values of zero precip should be left white in my opinion.*

While the authors agree that rainbow colour maps have been commonly used for precipitation diagrams, numerous recent studies have recommended against their use. General and hydrology specific comments on selecting appropriate colour maps are listed below. Additionally, we have consulted with a geospatial imagery specialist on the subject. As a result of extensively investigating and discussing the issue, we propose to follow the specialist advice recommending against rainbow colour maps.

https://ica-abs.copernicus.org/articles/1/96/2019/
https://www.sciencedirect.com/science/article/pii/S0924271622002659
https://hess.copernicus.org/articles/25/4549/2021/

An example of the reasoning is: "The disadvantages most often raised are as follows: (1) it is not perceptually ordered and there is no logical ordering (Monmonier 1991), which means there is no innate sense of higher or lower values; (2) it introduces sharp transitions between hues as the perceptual changes in the rainbow colours are not uniform (Moreland 2016), and which may be perceived as being a considerable transition in the mapped data; and (3) the middle values may be interpreted as extreme values since yellow has a highlighting effect, being perceived as brighter than the other colours." - https://doi.org/10.5194/ica-abs-1-96-2019

A more monotonic and appropriate rainbow colour map could be selected; however this introduces additional issues (e.g. similar colours but different colour scale to conventional rainfall rainbow colour maps).

In addition, the authors consider accessibility to be very important, and the selection of a rainbow colour has inclusivity issues, due to the fact that they are not colourblind safe. The authors are happy to consider other colour maps that are colour blind safe, and do not have mid-range values as white as this can also be difficult to understand, but on balance we and other readers are happy with the existing scheme.

5. *Equ 11: Fig. 20 shows that the effect of attenuation is maximal if R is large and the distance d to the radar is small, but MR(t) is largest if the distance is large and R large. As such, why is it used as an indicator for the effect shown in Fig 20. Shouldn't we choose an indicator that is inversely proportional to d ?*

While we agree that there may be an advantage in an indicator which co-varies with attenuation directly and could adapt the measure to be an inverse moment measure, we have found that this then puts a much greater emphasis on extremely large values, which is a very small subset of events. We feel that the existing measure is still intuitive to understand and based on the simplest available physical principles, and so prefer to leave as it is.

6. *Fig16 and 17: these figures show empty plots, without any explanation why!*

   The authors are confused as to which plots are empty, as online version we have access to has visible plots for Figure 16 and Figure 17. However, if there is an issue the authors have attached copies of these plots for clarification, and if there are any additional issues, please let us know.

7. *Fig21 and 22: : I don't get the difference between the two figures. It is not clear at all in the text. Is the second image for precipitation > 0.1 mmh−1 only?*

   We agree, thank you for raising this. From the captions it is not clear the difference between Figure 21 and Figure 22 and unclear in the text. Figure 21 shows the average ensemble behaviour, whilst Figure 22 show the ensemble variability (i.e. how much the RMSE or bias varies throughout the ensemble). The authors have clarified this in the revised manuscript, in both the captions and corresponding text, with a more concise Section 5.3 in general.

**Minor points to address:**

1. *l.21:I think you should also include the beam-broadening effect with non-uniform beam filling in the major error sources for QPE*
   The authors agree with this and have added this to the revised manuscript.

2. *l.52: Missing reference, should be Section 3*
   The authors agree with this and have added this to the revised manuscript.

3. *l.61: Since the study focuses on a single radar, it would be good to provide some info on this radar for people who are not familiar with the Met Office radar network (e.g. coordinates, frequency, PRF, pulse width)*
   The authors agree with this and have added this to the revised manuscript in Section 2.

4. *Equ3: please introduce the precipitation intensity variable R(t) properly*
5. The authors agree with this and have added this to the revised manuscript.

6. *Fig3: please indicate clearly in the caption that the blue histogram is empirical from radar data.*

The authors agree with this and have added this to the revised manuscript.

7.  *L133 eq8: c is not defined*

8.  The authors have rewritten Section 3.5, and have replaced Eq 8 with a more appropriate equation that better explains the methodology.

9.  *l.160: what is pixel variability? I imagine it is the variability between ensembles at pixel level. But this is defined and explained nowhere. The proportional error is also not defined explicitly.*

    The authors would like to thank reviewer 2 again for picking up on the missing information in the beginning of the results section. This has now been included, and rewritten to included more detail and explicit equations for uncertainty metrics in the revised manuscript.

10.  *Fig5: Please provide proper coordinates for your radar image, both in X and Y directions (-100 to 100 km). Do it at least in one image, you can then potentially indicate that, since the coordinates are always the same, you drop them from your figures. I saw that on Fig 7. 10 and 13, you indicated the coordinates, but starting from the top-left of your domain which is not standard. Please make everything more homogeneous.*

    The authors agree with this and have added proper coordinates to all relevant figures in the revised anuscript, and made all figures more homogeneous and consistend, and would like to thank the reviewer for pointing this out.

11.  *Fig12: the area of high precipitation is not visible in the image because the color scale of the rainfall field is not adequate. If the area is too small, use at least an arrow or a circle to show it. Also I wonder to which extent a precipitation field with such a tiny region of extremely high precipitation surrounded by almost zero everywhere else is realistic in a real world case (and is not caused by unfiltered clutter).*

    The authors agree with this and have highlighted the area of rainfall on the three figures for event B in the revised manuscript.

12.  *l.228: I don't understand proportion of rainfall in an images with the image RMSE, shouldn't it be proportion of heavy rainfall as in the plot y-label. And if yes, how is "heavy rainfall" defined? Is it the same as the significant rainfall from the PRS (1mmh−1 ), because in this case heavy is maybe a bit exaggerated?*

    The authors agree with this and have modified plot axes to explicitly show rainfall rates considered.

13.  *l.277: "imageS" (wrong capital S)*

    The authors agree with this and have corrected this in the revised manuscript.

14.  *Tab1: Please provide full name of ECDF here (empirical cumulative distribution function) as it the acronym is used before Fig23.*

15. The authors agree with this and have added this to the revised manuscript.

16. *The addition of letters (a, b, c, . . . ) in the captions of multiple panel figures (e.g. Fig 11, 22,...) would make it much easier for the reader to immediately link the corresponding plot to the reference in the caption.*
The authors agree with this and have added this to all relevant figures and text in the revised manuscript.

The authors agree with comments in the minor points and would like to thank the reviewers for these suggestions.

---

## Referee Report (RR1)

I would like to thank the authors for their careful and exhaustive review of the manuscript and for taking into account the remarks of the reviewers. I think that the article gained a lot in clarity and overall quality, in particular section 3.5. I am in favor of publishing it as is, with just a few very minor corrections.

l.253: please rephrase the last sentence, it is not clear.

Fig. 7: the colors in figures b,c, e and f) seem to be washed out (as if a value of alpha < 1 was specified).

Fig 8. and 11., given the range of reflectivity values I wonder whether it makes sense to reduce the vmax in the figure (< 60 dBZ) to improve the visibility. Regarding the colormaps I understand your position but still think that Viridis is not ideal, especially since custom colorblind friendly colormaps have been proposed by the radar community (see for example https://ntrs.nasa.gov/api/citations/20180004634/downloads/20180004634.pdf), but I leave it to author's appreciation.

l. 270. Should "were a as a result" be rather "resulted from" ? Please rephrase slightly, I found the sentence a bit clumsy.

l.274: is there a missing reference?

---

## Author Response (AR2)

I would like to thank the authors for their careful and exhaustive review of the manuscript and for taking into account the remarks of the reviewers. I think that the article gained a lot in clarity and overall quality, in particular section 3.5. I am in favor of publishing it as is, with just a few very minor corrections.

**The authors would like to thank the reviewers for all comments.**

l.253: please rephrase the last sentence, it is not clear.

**The sentence has been revised and clarified in the manuscript.**

Fig. 7: the colors in figures b,c, e and f) seem to be washed out (as if a value of alpha < 1 wasspecified).

**The slight variation between plots a, d and b, c, e, f differ slightly as the latter are plotted on a ploar grid using wradlib plotting software. The differing sizes of pixels results in a slight lighter colour where pixels become very dense towards the center of the images. Attempts were made to modify this however without reprojecting the data to a cartesian system (and losing information about the data when zooming in on the pdf) this slight difference in opacity around the edge of the pixels cannot be easily modified.**

Fig 8. and 11., given the range of reflectivity values I wonder whether it makes sense to reduce the vmax in the figure (< 60 dBZ) to improve the visibility. Regarding the colormaps I understand your position but still think that Viridis is not ideal, especially since custom colorblind friendly colormaps have been proposed by the radar community (see for example https://ntrs.nasa.gov/api/citations/20180004634/downloads/20180004634.pdf), but I leave it to author's appreciation.

**The authors considered alternative suggested colour schemes have decided to keep the original colour schemes, but appreciate that a lower vmax in Fig 8 and Fig 11 is appropriate. These have now been modified in the manuscript.**

l. 270. Should "were a as a result" be rather "resulted from" ? Please rephrase slightly, I found the sentence a bit clumsy.

**The sentenace has been rephrased and clarified in the manuscript.**

l.274: is there a missing reference?

**The reference has been added to the bibliography.**